# Fast Minimum-norm Adversarial Attacks through Adaptive Norm Constraints

**Maura Pintor**
University of Cagliari, Italy
Pluribus One, Italy
`maura.pintor@unica.it`

**Fabio Roli**
University of Cagliari, Italy
Pluribus One, Italy
`roli@unica.it`

**Wieland Brendel**
Tübingen AI Center,
University of Tübingen, Germany
`wieland.brendel@uni-tuebingen.de`

**Battista Biggio**
University of Cagliari, Italy
Pluribus One, Italy
`battista.biggio@unica.it`

## Abstract

Evaluating adversarial robustness amounts to finding the minimum perturbation needed to have an input sample misclassified. The inherent complexity of the underlying optimization requires current gradient-based attacks to be carefully tuned, initialized, and possibly executed for many computationally-demanding iterations, even if specialized to a given perturbation model. In this work, we overcome these limitations by proposing a fast minimum-norm (FMN) attack that works with different $\ell_p$-norm perturbation models ($p = 0, 1, 2, \infty$), is robust to hyperparameter choices, does not require adversarial starting points, and converges within few lightweight steps. It works by iteratively finding the sample misclassified with maximum confidence within an $\ell_p$-norm constraint of size $\epsilon$, while adapting $\epsilon$ to minimize the distance of the current sample to the decision boundary. Extensive experiments show that FMN significantly outperforms existing $\ell_0$, $\ell_1$, and $\ell_\infty$-norm attacks in terms of perturbation size, convergence speed and computation time, while reporting comparable performances with state-of-the-art $\ell_2$-norm attacks. Our open-source code is available at: `https://github.com/pralab/Fast-Minimum-Norm-FMN-Attack`.

## 1 Introduction

Learning algorithms are vulnerable to adversarial examples, i.e., intentionally-perturbed inputs aimed to mislead classification at test time [24, 3]. While adversarial examples have received much attention, evaluating the robustness of deep networks against them remains a challenge. Adversarial attacks solve a non-convex optimization problem and are thus prone to finding suboptimal solutions; in particular, all attacks make certain assumptions about the underlying geometry and properties of the optimization problem which, if violated, can derail the attack and may lead to premature conclusions regarding model robustness. That is why the vast majority of defenses published in recent years have later shown to be ineffective against more powerful white-box attacks [5, 1]. Having an arsenal of diverse attacks that can be adapted to specific defenses is one of the most promising avenues for increasing confidence in white-box robustness evaluations [6, 25]. While it may seem that the number of attacks is already large, most of them are just small variations of the same technique, make similar underlying assumptions and thus tend to fail jointly (see, e.g., [25], in which projected-gradient attacks all fail similarly against the "Ensemble Diversity" defense).

35th Conference on Neural Information Processing Systems (NeurIPS 2021).

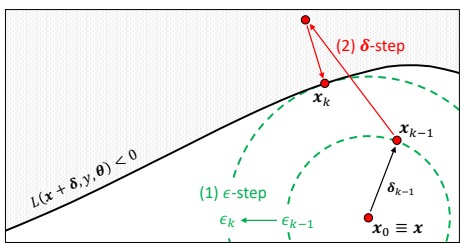 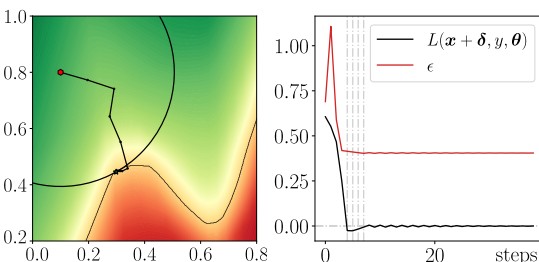

Figure 1: (a) Conceptual representation of the FMN attack algorithm (*leftmost plot*). The $\epsilon$-step updates the constraint size $\epsilon$ to minimize its distance to the boundary. The $\boldsymbol{\delta}$-step updates the perturbation $\boldsymbol{\delta}$ with a projected-gradient step to maximize misclassification confidence within the current $\epsilon$-sized constraint. (b) Example of execution of our attack on a bi-dimensional problem (*middle plot*), along with the corresponding values of the loss function $L$ and the constraint size $\epsilon$ across iterations (*rightmost plot*). Our algorithm works by first pushing the initial point (red dot) towards the adversarial region (in red), and then perturbing it around the decision boundary to improve the current solution towards a local optimum. The vertical lines in the rightmost plot highlight the steps in which a better solution (smaller $\|\boldsymbol{\delta}^\star\|$ and $L < 0$) is found.

In this work, we focus on *minimum-norm* attacks for evaluating adversarial robustness, i.e., attacks that aim to mislead classification by finding the smallest input perturbation according to a given norm. In contrast to *maximum-confidence* attacks, which maximize confidence in a wrong class within a given perturbation budget, the former are better suited to evaluate adversarial robustness as one can compute the accuracy of a classifier under attack for any perturbation budget without re-running the attack. Within the class of gradient-based minimum-norm attacks, there are three main sub-categories: (i) soft-constraint attacks, (ii) boundary attacks and (iii) projected-gradient attacks. Soft-constraint attacks like CW [5] optimize a trade-off between confidence of the misclassified samples and perturbation size. This class of attacks needs a sample-wise tuning of the trade-off hyperparameter to find the smallest possible perturbation, thus requiring many steps to converge. Boundary attacks like BB [4] and FAB [10] move along the decision boundary towards the closest point to the input sample. These attacks converge within relatively few steps, but BB requires an adversarial starting point, and both attacks need to solve a relatively expensive optimization problem in each step. Finally, recent minimum-norm projected-gradient attacks like DDN [23] perform a maximum-confidence attack in each step under a given perturbation budget $\epsilon$, while iteratively adjusting $\epsilon$ to reduce the perturbation size. DDN combines the effectiveness of boundary attacks with the simplicity and per-step speed of soft-constraint attacks; however, it is specific to the $\ell_2$ norm and cannot be readily extended to other norms.

To overcome the aforementioned limitations, in this work we propose a novel, fast minimum-norm (FMN) attack (Sect. 2), which retains the main advantages of DDN while generalizing it to different $\ell_p$ norms ($p = 0, 1, 2, \infty$). We perform large-scale experiments on different datasets and models (Sect. 3), showing that FMN is able to significantly outperform current minimum-norm attacks in terms of convergence speed and computation time (except for $\ell_2$-norm attacks, for which FMN achieves comparable results), while finding equal or better optima, on average, across almost all tested scenarios and $\ell_p$ norms. FMN thus combines all desirable traits a good adversarial attack should have, providing an important step towards improving adversarial robustness evaluations. We conclude the paper by discussing related work (Sect. 4) and future research directions (Sect. 5).

## 2 Minimum-Norm Adversarial Examples with Adaptive Projections

**Problem formulation.** Given an input sample $\boldsymbol{x} \in [0,1]^d$, belonging to class $y \in \{1, \ldots, c\}$, the goal of an untargeted attack is to find the minimum-norm perturbation $\boldsymbol{\delta}^\star$ such that the corresponding adversarial example $\boldsymbol{x}^\star = \boldsymbol{x} + \boldsymbol{\delta}^\star$ is misclassified. This problem can be formulated as:

$$\boldsymbol{\delta}^\star \in \arg\min_{\boldsymbol{\delta}} \quad \|\boldsymbol{\delta}\|_p \,, \tag{1}$$

$$\text{s.t.} \quad L(\boldsymbol{x} + \boldsymbol{\delta}, y, \boldsymbol{\theta}) < 0 \,, \tag{2}$$

$$\boldsymbol{x} + \boldsymbol{\delta} \in [0,1]^d \,, \tag{3}$$

---

**Algorithm 1** Fast Minimum-norm (FMN) Attack

---

**Input:** $\boldsymbol{x}$, the input sample; $t$, a variable denoting whether the attack is targeted ($t = +1$) or untargeted ($t = -1$); $y$, the target (true) class label if the attack is targeted (untargeted); $\gamma_0$ and $\gamma_K$, the initial and final $\epsilon$-step sizes; $\alpha_0$ and $\alpha_K$, the initial and final $\boldsymbol{\delta}$-step sizes; $K$, the total number of iterations.

**Output:** The minimum-norm adversarial example $\boldsymbol{x}^\star$.

1: $\boldsymbol{x}_0 \leftarrow \boldsymbol{x}$, $\epsilon_0 = 0$, $\boldsymbol{\delta}_0 \leftarrow \boldsymbol{0}$, $\boldsymbol{\delta}^\star \leftarrow \infty$
2: **for** $k = 1, \ldots, K$ **do**
3:    $\boldsymbol{g} \leftarrow t \cdot \nabla_{\boldsymbol{\delta}} L(\boldsymbol{x}_{k-1} + \boldsymbol{\delta}, y, \boldsymbol{\theta})$   *// loss gradient*
4:    $\gamma_k \leftarrow h(\gamma_0, \gamma_K, k, K)$   *// $\epsilon$-step size decay* (Eq. 6)
5:    **if** $L(\boldsymbol{x}_{k-1}, y, \boldsymbol{\theta}) \geq 0$ **then**
6:       $\epsilon_k = \|\boldsymbol{\delta}_{k-1}\|_p + L(\boldsymbol{x}_{k-1}, y, \boldsymbol{\theta})/\|\boldsymbol{g}\|_q$ **if** *adversarial not found yet* **else** $\epsilon_k = \epsilon_{k-1}(1 + \gamma_k)$
7:    **else**
8:       **if** $\|\boldsymbol{\delta}_{k-1}\|_p \leq \|\boldsymbol{\delta}^\star\|_p$ **then**
9:          $\boldsymbol{\delta}^\star \leftarrow \boldsymbol{\delta}_{k-1}$   *// update best min-norm solution*
10:      **end if**
11:      $\epsilon_k = \min(\epsilon_{k-1}(1 - \gamma_k), \|\boldsymbol{\delta}^\star\|_p)$
12:    **end if**
13:    $\alpha_k \leftarrow h(\alpha_0, \alpha_K, k, K)$   *// $\boldsymbol{\delta}$-step size decay* (Eq. 6)
14:    $\boldsymbol{\delta}_k \leftarrow \boldsymbol{\delta}_{k-1} + \alpha_k \cdot \boldsymbol{g}/\|\boldsymbol{g}\|_2$   *// gradient-scaling step*
15:    $\boldsymbol{\delta}_k \leftarrow \Pi_\epsilon(\boldsymbol{x}_0 + \boldsymbol{\delta}_k) - \boldsymbol{x}_0$
16:    $\boldsymbol{\delta}_k \leftarrow \text{clip}(\boldsymbol{x}_0 + \boldsymbol{\delta}_k) - \boldsymbol{x}_0$
17:    $\boldsymbol{x}_k \leftarrow \boldsymbol{x}_0 + \boldsymbol{\delta}_k$
18: **end for**
19: **return** $\boldsymbol{x}^\star \leftarrow \boldsymbol{x}_0 + \boldsymbol{\delta}^\star$

---

where $|| \cdot ||_p$ indicates the $\ell_p$-norm operator. The loss $L$ in the constraint in Eq. (2) is defined as:

$$L(\boldsymbol{x}, y, \boldsymbol{\theta}) = f_y(\boldsymbol{x}, \boldsymbol{\theta}) - \max_{j \neq y} f_j(\boldsymbol{x}, \boldsymbol{\theta}), \tag{4}$$

where $f_j(\boldsymbol{x}, \boldsymbol{\theta})$ is the confidence given by the model $f$ for classifying $\boldsymbol{x}$ as class $j$, and $\boldsymbol{\theta}$ is the set of its learned parameters. Assuming that the classifier assigns $\boldsymbol{x}$ to the class exhibiting the highest confidence, i.e., $y^\star = \arg\max_{j \in 1,\ldots,c} f_j(\boldsymbol{x}, \boldsymbol{\theta})$, the loss function $L(\boldsymbol{x}, y, \boldsymbol{\theta})$ takes on negative values only when $\boldsymbol{x}$ is misclassified. Finally, the box constraint in Eq. (3) ensures that the perturbed sample $\boldsymbol{x} + \boldsymbol{\delta}$ lies in the feasible input space. The aforementioned problem typically involves a non-convex loss function $L$ (w.r.t. its first argument), due to the non-convexity of the underlying decision function $f$. For this reason, it may admit different locally-optimal solutions. Note also that the solution is trivial (i.e., $\boldsymbol{\delta}^\star = \boldsymbol{0}$) when the input sample $\boldsymbol{x}$ is already adversarial (i.e., $L(\boldsymbol{x}, y, \boldsymbol{\theta}) < 0$).

**Extension to the targeted case.** The goal of a targeted attack is to have the input sample misclassified in a given target class $y'$. This can be accounted for by modifying the loss function in Eq. (4) as $L^t(\boldsymbol{x}, y', \boldsymbol{\theta}) = \max_{j \neq y'} f_j(\boldsymbol{x}, \boldsymbol{\theta}) - f_{y'}(\boldsymbol{x}, \boldsymbol{\theta}) = -L(\boldsymbol{x}, y', \boldsymbol{\theta})$, i.e., changing its sign and using the target class label $y'$ instead of the true class label $y$.

**Solution algorithm.** To solve Problem (1)-(3), we reformulate it using an upper bound $\epsilon$ on $\|\boldsymbol{\delta}\|_p$:

$$\min_{\epsilon, \boldsymbol{\delta}} \epsilon, \quad \text{s.t. } \|\boldsymbol{\delta}\|_p \leq \epsilon, \tag{5}$$

and to the constraints in Eqs. (2)-(3). This allows us to derive an algorithm that works in two main steps, similarly to DDN [23], by updating the maximum perturbation size $\epsilon$ separately from the actual perturbation $\boldsymbol{\delta}$, as represented in Fig. 1(a). In particular, the constraint size $\epsilon$ is adapted to reduce the distance of the constraint to the boundary ($\epsilon$-step), while the perturbation $\boldsymbol{\delta}$ is updated using a projected-gradient step to minimize the loss function $L$ within the given $\epsilon$-sized constraint ($\boldsymbol{\delta}$-step). This essentially amounts to a projected gradient descent algorithm that iteratively adapts the constraint size $\epsilon$ to find the minimum-norm adversarial example. The complete algorithm is given as Algorithm 1, while a more detailed explanation of the two aforementioned steps is given below.

$\epsilon$-**step.** This step updates the upper bound $\epsilon$ on the perturbation norm (lines 4-12 in Algorithm 1). The underlying idea is to increase $\epsilon$ if the current sample is not adversarial (i.e., $L(\boldsymbol{x}_{k-1}, y, \boldsymbol{\theta}) \geq 0$), and to decrease it otherwise, while reducing the step size to dampen oscillations around the boundary

and reach convergence. In the former case ($\epsilon$-*increase*), the increment of $\epsilon$ depends on whether an adversarial example has been previously found or not. If not, we estimate the distance to the boundary with a first-order linear approximation, and set $\epsilon_k = \|\boldsymbol{\delta}_{k-1}\|_p + L(\boldsymbol{x}_{k-1}, y, \boldsymbol{\theta})/\|\nabla L(\boldsymbol{x}_{k-1}, y, \boldsymbol{\theta})\|_q$, where $q$ is the dual norm of $p$. This approximation allows the attack point to make faster progress towards the decision boundary. Conversely, if an adversarial sample has been previously found, but the current sample is not adversarial, it is likely that the current estimate of $\epsilon$ is only slightly smaller than the minimum-norm solution. We thus increase $\epsilon$ by a small fraction as $\epsilon_k = \epsilon_{k-1}(1 + \gamma_k)$, being $\gamma_k$ a decaying step size. In the latter case ($\epsilon$-*decrease*), if the current sample is adversarial, i.e., $L(\boldsymbol{x}_{k-1}, y, \boldsymbol{\theta}) < 0$, we decrease $\epsilon$ as $\epsilon_k = \epsilon_{k-1}(1 - \gamma_k)$, to check whether the current solution can be improved. If the corresponding $\epsilon_k$ value is larger than the optimal $\|\boldsymbol{\delta}^\star\|_p$ found so far, we retain the best value and set $\epsilon_k = \|\boldsymbol{\delta}^\star\|_p$. These multiplicative updates of $\epsilon$ exhibit an oscillating behavior around the decision boundary, due to the conflicting requirements of minimizing the perturbation size and finding an adversarial point. To ensure convergence, as anticipated before, the step size $\gamma_k$ is decayed with cosine annealing:

$$\gamma_k = h(\gamma_0, \gamma_K, k, K) = \gamma_K + \tfrac{1}{2}(\gamma_0 - \gamma_K)\left(1 + \cos\left(\tfrac{k\pi}{K}\right)\right) , \tag{6}$$

being $k$ the current step, $K$ the total number of steps, and $\gamma_0$ and $\gamma_K$ the initial and final step sizes.

$\boldsymbol{\delta}$**-step.** This step updates $\boldsymbol{\delta}$ (lines 13-17 in Algorithm 1). The goal is to find the adversarial example that is misclassified with maximum confidence (i.e., for which $L$ is minimized) within the current $\epsilon$-sized constraint (Eq. 5) and bounds (Eq. 3). This amounts to performing a projected-gradient step along the negative gradient of $L$. We consider a normalized steepest descent with decaying step size $\alpha$ to overcome potential issues related to noisy gradients while ensuring convergence (line 14). Note that this step only rescales the gradient by its $\ell_2$ norm, while preserving its direction. The step size $\alpha$ is decayed using cosine annealing (Eq. 6). Once $\boldsymbol{\delta}$ is updated, we project it onto the given $\epsilon$-sized $\ell_p$-norm constraint via a projection operator $\Pi_\epsilon$ (line 15), to fulfill the constraint in Eq. (5). The projection is trivial for $p = \infty$ and $p = 2$. For $p = 1$, we use the efficient algorithm by Duchi et al. [14]. For $p = 0$, we retain only the first $\epsilon$ components of $\boldsymbol{\delta}$ exhibiting the largest absolute value. We finally clip the components of $\boldsymbol{\delta}$ that violate the bounds in Eq. (3) (line 16).

**Execution example.** In Fig. 1(b), we report an example of execution of our algorithm on a bi-dimensional problem. The initial sample is updated to follow the negative gradient of $L$ towards the decision boundary. When an adversarial point is found, the algorithm reduces $\epsilon$ to find a better solution. The point is thus projected back onto the non-adversarial region, and $\epsilon$ increased (by a smaller, decaying amount). These oscillations allow the point to walk on the boundary towards a local optimum, i.e., an adversarial point lying on the boundary, where the gradient of the loss function and that of the norm constraint have opposite direction. FMN tends to quickly converge to a good local optimum, provided that the step size is reduced to a sufficiently-small value and that a sufficiently-large number of iterations are performed. This is also confirmed empirically in Sect. 3.

**Adversarial initialization.** Our attack can be initialized from the input sample $\boldsymbol{x}$, or from a point $\boldsymbol{x}_{\text{init}}$ belonging either to a different class (if the attack is untargeted) or to the target class (if the attack is targeted). When initializing the attack from $\boldsymbol{x}_{\text{init}}$, we perform a 10-step binary search between $\boldsymbol{x}$ and $\boldsymbol{x}_{\text{init}}$, to find an adversarial point which is closer to the decision boundary. In particular, we aim to find the minimum $\epsilon$ such that $L(\boldsymbol{x} + \Pi_\epsilon(\boldsymbol{x}_{\text{init}} - \boldsymbol{x}), y, \boldsymbol{\theta}) < 0$ (or $L^t < 0$ for targeted attacks). Then we run our attack starting from the corresponding values of $\boldsymbol{x}_k$, $\epsilon_k$, $\boldsymbol{\delta}_k$ and $\boldsymbol{\delta}^\star$.

**Differences with DDN.** FMN applies substantial changes to both the algorithm and the formulation of DDN. The main difference is that (i) DDN always rescales the perturbation to have size $\epsilon$. This operation is problematic when using other norms, especially sparse ones, as it hinders the ability of the attack to explore the neighboring space and find a suitable descent direction. Another difference is that (ii) FMN does not use the cross-entropy loss, but it uses the logit difference as the loss function $L$, since the latter is less affected by saturation effects. Moreover, (iii) FMN does not need an initial value for $\epsilon$, as $\epsilon$ is dynamically estimated; and (iv) $\gamma$ is decayed to improve convergence around better minimum-norm solutions, by more effectively dampening oscillations around the boundary. Finally, we include the possibility of (v) initializing the attack from an adversarial point, which can greatly increase the convergence speed of the algorithm, as it uses a fast line-search algorithm to find the boundary and the remaining queries to refine the result.

# 3  Experiments

We report here an extensive experimental analysis involving several state-of-the-art defenses and minimum-norm attacks, covering $\ell_0$, $\ell_1$, $\ell_2$ and $\ell_\infty$ norms. The goal is to empirically benchmark our attack and assess its effectiveness and efficiency as a tool for adversarial robustness evaluation.

## 3.1  Experimental Setup

**Datasets.** We consider two commonly-used datasets for benchmarking adversarial robustness of deep neural networks, i.e., the MNIST handwritten digits and CIFAR10. Following the experimental setup in [4], we use a subset of 1000 test samples to evaluate the considered attacks and defenses.

**Models.** We use a diverse selection of models to thoroughly evaluate attacks under different conditions. For MNIST, we consider the following four models: *M1*, the 9-layer network used as the undefended baseline model by Papernot et al. [20], Carlini and Wagner [5]; *M2*, the robust model by Madry et al. [17], trained on $\ell_\infty$ attacks (robustness claim: 89.6% accuracy with $\|\boldsymbol{\delta}\|_\infty \leq 0.3$, current best evaluation: 88.0%); *M3*, the robust model by Rony et al. [23], trained on $\ell_2$ attacks (robustness claim: 87.6% accuracy with $\|\boldsymbol{\delta}\|_2 \leq 1.5$); and *M4*, the IBP Large Model by Zhang et al. [27] (robustness claim: 94.3% accuracy with $\|\boldsymbol{\delta}\|_\infty \leq 0.3$). For CIFAR10, we consider three state-of-the-art robust models from RobustBench [11]: *C1*, the robust model by Madry et al. [17], trained on $\ell_\infty$ attacks (robustness claim: 44.7% accuracy with $\|\boldsymbol{\delta}\|_\infty \leq 8/255$, current best evaluation: 44.0%); *C2*, the defended model by Carmon et al. [7] (top-5 in RobustBench), trained on $\ell_\infty$ attacks and additional unsupervised data (robustness claim: 62.5% accuracy with $\|\boldsymbol{\delta}\|_\infty \leq 8/255$, current best evaluation: 59.5%); and *C3*, the robust model by Rony et al. [23], trained on $\ell_2$ attacks (robustness claim: 67.9% accuracy with $\|\boldsymbol{\delta}\|_2 \leq 0.5$, current best evaluation: 66.4%).

**Attacks.** We compare our algorithm against different state-of-the-art attacks for finding minimum-norm adversarial perturbations across different norms: the Carlini & Wagner (CW) attack [5], the Decoupling Direction and Norm (DDN) attack [23], the Brendel & Bethge (BB) attack [4], and the Fast Adaptive Boundary (FAB) attack [10]. We use the implementation of FAB from Ding et al. [12], while for all the remaining attacks we use the implementation available in Foolbox [21, 22]. All these attacks are defined on the $\ell_2$ norm. BB and FAB are also defined on the $\ell_1$ and $\ell_\infty$ norms, and only BB is defined on the $\ell_0$ norm. We consider both untargeted and targeted attack scenarios, as defined in Sect. 2, except for FAB, which is only evaluated in the untargeted case.[1]

**Hyperparameters.** To ensure a fair comparison, we perform an extensive hyperparameter search for each of the considered attacks. We consider two main scenarios: tuning the hyperparameters at the *sample-level* and at the *dataset-level*. In the sample-level scenario, we select the optimal hyperparameters separately for each input sample by running each attack 10 to 16 times per sample, with a different hyperparameter configuration or random initialization point each time. In the dataset-level scenario, we choose the same hyperparameters for all samples, selecting the configuration that yields the best attack performance. While sample-level tuning provides a fairer comparison across attacks, it is more computationally demanding and less practical than dataset-level tuning. In addition, the latter allows us to understand how robust attacks are to suboptimal hyperparameter choices. We select the hyperparameters to be optimized for each attack as recommended by the corresponding authors [4, 5, 10, 23]. The hyperparameter configurations considered for each attack are detailed below. For attacks that are claimed to be robust to hyperparameter changes, like BB and FAB, we follow the recommendation of using a larger number of random restarts rather than increasing the number of hyperparameter configurations to be tested. In addition, as BB requires being initialized from an adversarial starting point, we initialize it by randomly selecting a sample either from a different class (in the untargeted case) or from the target class (in the targeted case). Finally, as each attack performs operations with different levels of complexity within each iteration, possibly querying the model multiple times, we set the number of steps for each attack such that at least $1,000$ forward passes (i.e., *queries*) are performed. This ensures a fairer comparison also in terms of the computational time and resources required to execute each attack.

---

[1]The reason is that FAB does not support generating adversarial examples with a particular target label. The targeted version of FAB aims to find a closer untargeted misclassification by running the attack a number of times, each time targeting a different candidate class, and then selecting the best solution [10, 9].

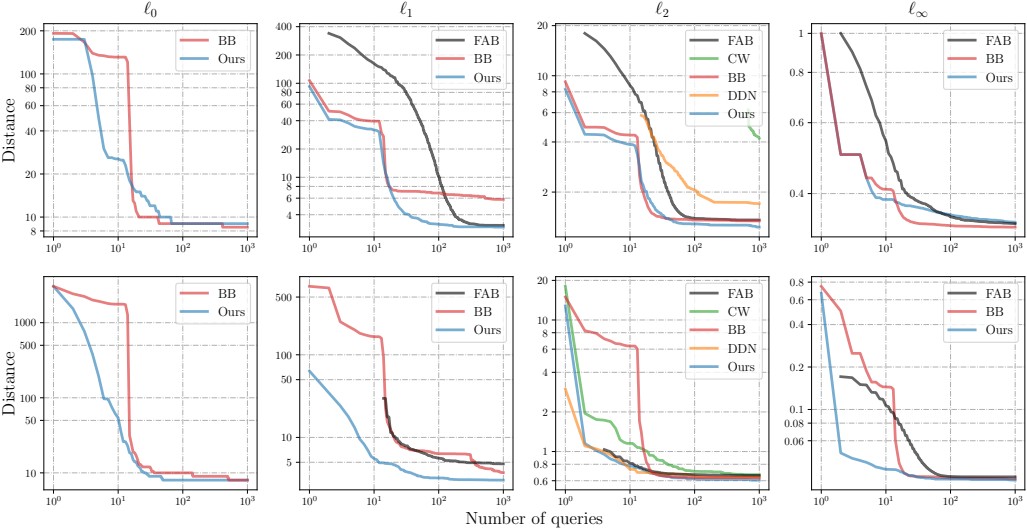

Figure 2: Query-distortion curves for MNIST (M2, *top*) and CIFAR10 (C1, *bottom*) models (untargeted scenario).

*CW.* This attack minimizes the soft-constraint version of our problem, i.e., $\min_{\boldsymbol{\delta}} \|\boldsymbol{\delta}\|_p + c \cdot \min(L(\boldsymbol{x} + \boldsymbol{\delta}, y, \boldsymbol{\theta}), -\kappa)$. The hyperparameters $\kappa$ and $c$ are used to tune the trade-off between perturbation size and misclassification confidence. To find minimum-norm perturbations, CW requires setting $\kappa = 0$, while the constant $c$ is tuned via binary search (re-running the attack at each iteration). We set the number of binary-search steps to 9, and the maximum number of iterations to 250, to ensure that at least $1,000$ queries are performed. We also set different values for $c, \eta \in \{10^{-3}, 10^{-2}, 10^{-1}, 1\}$.

*DDN.* This attack, similarly to ours, maximizes the misclassification confidence within an $\epsilon$-sized constraint, while adjusting $\epsilon$ to minimize the perturbation size. We consider initial values of $\epsilon_0 \in \{0.03, 0.1, 0.3, 1, 3\}$, and run the attack with a different number of iterations $K \in \{200, 1000\}$, as this affects the size of each update on $\boldsymbol{\delta}$.

*BB.* This attack starts from a randomly-drawn adversarial point, performs a 10-step binary search to find a point which is closer to the decision boundary, and then updates the point to minimize its perturbation size by following the decision boundary. In each iteration, BB computes the optimal update within a given trust region of radius $\rho$. We consider different values for $\rho \in \{10^{-3}, 10^{-2}, 10^{-1}, 1\}$, while we fix the number of steps to 1000. We run the attack 3 times by considering different initialization points, and eventually retain the best solution.

*FAB.* This attack iteratively optimizes the attack point by linearly approximating its distance to the decision boundary. It uses an adaptive step size bounded by $\alpha_{\max}$ and an extrapolation step $\eta$ to facilitate finding adversarial points. As suggested by Croce and Hein [10], we tune $\alpha_{\max} \in \{0.1, 0.05\}$ and $\eta \in \{1.05, 1, 3\}$. We consider 3 different random initialization points, and run the attack for 500 steps each time, eventually selecting the best solution.

*FMN.* We run FMN for $K = 1000$ steps, using $\gamma_0 \in \{0.05, 0.3\}$, $\gamma_K = 10^{-4}$, and $\alpha_K = 10^{-5}$. For $\ell_0$, $\ell_1$, and $\ell_2$, we set $\alpha_0 \in \{1, 5, 10\}$. For $\ell_\infty$, we set $\alpha_0 \in \{10^1, 10^2, 10^3\}$, as the normalized $\ell_2$ step yields much smaller updates in the $\ell_\infty$ norm. For each hyperparameter setting we run the attack twice, starting from (i) the input sample and (ii) an adversarial point.

**Evaluation criteria.** We evaluate the attacks along four different criteria: (i) *perturbation size* and (ii) *robustness to hyperparameter selection*, measured as the median $\|\boldsymbol{\delta}^\star\|_p$ on the test set (for a fixed budget of $Q$ queries and for sample- and dataset-level hyperparameter tuning, where by "robustness" we mean that a fixed hyperparameter configuration works well across different samples); (iii) *execution time*, measured as the average time spent per query (in milliseconds); and (iv) *convergence speed*, measured as the average number of queries required to converge to a good-enough solution (within 10% of the best value found at $Q = 1000$). When computing the median, we follow the evaluation in [4]: the perturbation size is set to 0 if a clean sample is misclassified, while it is set

to $\infty$ when the attack fails (no adversarial is found). The median perturbation size thus represents the value for which 50% of the samples evade a particular model.

## 3.2 Experimental Results

*Query-distortion (QD) curves.* To evaluate each attack in terms of perturbation size under the same query budget $Q$, we use the so-called QD curves introduced by Brendel et al. [4]. These curves report, for each attack, the median value of $\delta^\star$ as a function of the number of queries $Q$. For each given $Q$ value, the optimal $\delta^\star$ for each point is selected among the different attack executions (i.e., using different hyperparameters and/or initialization points, as described in Sect. 3.1). In Fig. 2, we report the QD curves for the MNIST and CIFAR10 challenge models (i.e., M2 and C1) in the untargeted scenario. The remaining QD curves exhibit a similar behavior and can be found in the supplementary material. It is worth noting that our attack attains comparable results in terms of perturbation size across all norms, while significantly outperforming FAB and BB in the $\ell_1$ case. It typically requires also less iterations than the other attacks to converge. While the QD curves show the complete behavior of each attack as $Q$ increases, a more compact and thorough summary of our evaluation is reported below, according to the four evaluation criteria described in Sect. 3.1.

**Perturbation size.** Table 1 reports the median value of $\|\delta^\star\|$ at $Q = 1000$ queries (i.e., the last value from the query-distortion curve), for all models, attacks and norms. The values obtained with sample-level hyperparamter tuning confirm that our attack can find smaller or comparable perturbations with those found by the competing attacks, in most of the untargeted and targeted cases, and that the biggest margin is achieved in the $\ell_1$ case. FMN is only slightly worse than DDN and BB in a few cases, including $\ell_2$-DDN on M4 and $\ell_\infty$-BB on M2 and M4. The reason may be that these robust models exhibit noisy gradients and flat regions around the clean input samples, hindering the initial optimization steps of the FMN attack.

**Robustness to hyperparameter selection.** The values reported in the lower part of Table 1 show that, when using dataset-level hyperparameter tuning, FMN outperforms the other attacks in a much larger number of cases. This shows that FMN is more robust to hyperparameter changes, while other attacks like $\ell_0$- and $\ell_1$-BB suffer when using the same hyperparameters for all samples.

**Execution time.** The average runtime per query for each attack-model pair, measured on a workstation with an NVIDIA GeForce RTX 2080 Ti GPU with 11GB of RAM, can be found in Table 2. The results show that our attack is up to 2-3 times faster, with the exception of DDN in the $\ell_2$ case. This is however compensated by the fact that FMN finds better solutions. The advantage is that our attack avoids costly inner projections as in BB and FAB. FMN is slightly less time-efficient than DDN and CW, as it simultaneously updates the adversarial point and the norm constraint. In particular, the update on the constraint may initially require computing the norm of the gradient $g$ (line 6 in Algorithm 1), which increases the runtime of our attack. FAB computes a similar step, but for all the output classes, which hinders its scalability to problems with many classes.

**Convergence speed.** To get an estimate of the convergence speed, we measure the number of queries required by each attack to reach a perturbation size that is within 10% of the value found at $Q = 1000$ queries (the lower the better). Results are shown in Table 3. Our attack converges on par with or faster than all other attacks for almost all models, often requiring only half or a fifth as many queries as the state of the art. Exceptions are MNIST and CIFAR10 challenge models (M2 and C1) for $\ell_2$ and $\ell_\infty$, where BB and DDN occasionally converge faster. FMN rarely needs more than 100 steps, reaching the minimal perturbation after only 10-30 queries on many datasets, models and norms.

*Robust accuracy.* Despite our attack being not tailored to target specific defenses, and our evaluation restricted to a subset of the testing samples, it is worth remarking that the robust accuracies of the models against our attack are aligned with that reported in current evaluations, with the notable exception of C3, where our attack can decrease robust accuracy from 67.9% to 65.5%.

**Experiments on ImageNet.** We conclude our experiments by running an additional comparison between FMN and a widely-used maximum-confidence attack, i.e., the Projected Gradient Descent (PGD) attack [17], on two pretrained ImageNet models (i.e., ResNet18 and VGG16), considering $\ell_1$, $\ell_2$ and $\ell_\infty$ norms. The hyperparameters are tuned at the *dataset-level* using 20 validation samples. For FMN, we fix the hyperparameters as discussed before, and only tune $\alpha_0 \in \{0.1, 1, 2, 8\}$, without using adversarial initialization. For PGD, we tune the step size $\alpha \in \{0.001, 0.01, 0.1, 1, 2, 8\}$. We run both attacks for $Q = 1,000$ queries on a separate set of $1,000$ samples. The success rates of both

Table 1: Median $\|\boldsymbol{\delta}^\star\|_p$ value at $Q = 1000$ queries for targeted and untargeted attacks, with sample-level and dataset-level hyperparameter tuning.

| | | MNIST | | | | | | | | CIFAR10 | | | | | |
|---|---|---|---|---|---|---|---|---|---|---|---|---|---|---|---|
| | | *Untargeted* | | | | *Targeted* | | | | *Untargeted* | | | *Targeted* | | |
| | Model | M1 | M2 | M3 | M4 | M1 | M2 | M3 | M4 | C1 | C2 | C3 | C1 | C2 | C3 |
| | | *Sample-level Hyperparameter Tuning* | | | | | | | | | | | | | |
| $\ell_0$ | BB | **7** | **8** | **15** | 94 | **14** | 27 | **24** | 93 | **8** | 12 | **13** | **19** | **32** | 25 |
| | Ours | **7** | 9 | **15** | 5 | **14** | 20 | 24 | 23 | **8** | 11 | 14 | **19** | **32** | 27 |
| $\ell_1$ | FAB | 6.60 | 3.08 | 14.23 | 109.4 | - | - | - | **6.25** | 4.79 | 5.17 | 8.79 | - | - | - |
| | BB | 6.26 | 5.81 | 13.16 | 5.44 | 12.42 | 10.38 | 20.41 | **6.25** | 3.75 | 4.29 | 8.62 | 8.04 | 10.93 | 15.71 |
| | Ours | **5.57** | **2.95** | **12.04** | **1.96** | **12.20** | **6.75** | **18.79** | 7.31 | **3.04** | **3.43** | **8.26** | **7.07** | **9.40** | **15.24** |
| $\ell_2$ | FAB | 1.45 | 1.36 | 2.62 | 2.97 | - | - | - | - | 0.66 | 0.72 | 0.94 | - | - | - |
| | CW | 1.49 | 4.22 | 2.78 | - | 2.33 | 6.97 | 3.54 | - | 0.67 | 0.74 | **0.91** | 1.08 | 1.27 | 1.38 |
| | BB | 1.43 | 1.34 | 2.61 | 1.61 | **2.27** | 2.04 | 3.23 | 1.79 | 0.63 | 0.70 | **0.91** | 1.07 | 1.26 | 1.38 |
| | DDN | 1.46 | 1.71 | 2.56 | **0.79** | 2.29 | 2.20 | 3.27 | **1.33** | 0.64 | 0.73 | **0.91** | 1.09 | 1.29 | 1.39 |
| | Ours | **1.41** | **1.23** | **2.50** | 0.94 | 2.28 | **1.89** | **3.19** | 1.85 | **0.61** | **0.69** | **0.91** | **1.03** | **1.21** | **1.38** |
| $\ell_\infty$ | FAB | .138 | .337 | .233 | .421 | - | - | - | - | .033 | .043 | .025 | - | - | - |
| | BB | .138 | **.330** | .227 | **.402** | .202 | **.355** | **.271** | **.403** | .032 | .041 | **.024** | **.055** | .064 | **.037** |
| | Ours | **.134** | .339 | **.226** | .404 | **.201** | .389 | .272 | .406 | **.032** | **.040** | **.024** | **.055** | **.063** | **.037** |
| | | *Dataset-level Hyperparameter Tuning* | | | | | | | | | | | | | |
| $\ell_0$ | BB | 12 | 152 | 52 | 145 | 20 | 179 | 39 | 183 | 28 | 44 | 32 | 29 | 65 | 33 |
| | Ours | **9** | **33** | **18** | **15** | **16** | **48** | **28** | **55** | **11** | **17** | **16** | **25** | **38** | **32** |
| $\ell_1$ | FAB | 8.66 | 225.7 | 163.9 | 312.3 | - | - | - | - | - | - | 20.48 | - | - | - |
| | BB | 10.60 | 49.83 | 17.57 | 46.99 | 16.60 | 53.11 | 29.89 | 54.31 | 7.02 | 10.20 | 17.13 | 11.41 | 15.26 | 23.37 |
| | Ours | **7.13** | **4.18** | **13.66** | **4.99** | **13.18** | **8.33** | **21.37** | **12.16** | **4.28** | **4.82** | **9.52** | **8.51** | **10.40** | **17.32** |
| $\ell_2$ | FAB | 1.54 | 1.59 | 2.81 | 16.30 | - | - | - | - | 0.77 | 1.11 | 1.06 | - | - | - |
| | CW | 1.63 | 5.15 | 3.71 | - | 2.50 | - | 4.72 | - | 0.86 | 1.00 | 0.99 | 1.36 | 2.90 | 1.55 |
| | BB | 1.75 | 1.82 | 3.02 | 4.57 | 2.64 | 2.59 | 3.52 | 5.31 | 0.86 | 0.95 | 1.10 | 1.25 | 1.45 | 1.73 |
| | DDN | **1.47** | 2.01 | 2.62 | **1.15** | 2.31 | 2.72 | 3.36 | **1.96** | **0.66** | 0.77 | 0.91 | 1.11 | 1.31 | 1.40 |
| | Ours | 1.61 | **1.42** | **2.61** | 1.56 | **2.30** | **2.13** | **3.24** | 2.41 | 0.67 | **0.74** | **0.91** | **1.09** | **1.28** | **1.38** |
| $\ell_\infty$ | FAB | .148 | .365 | .248 | .900 | - | - | - | - | .038 | .052 | .029 | - | - | - |
| | BB | .159 | **.336** | .243 | .409 | .223 | **.361** | .280 | .477 | .044 | .054 | .029 | .059 | .074 | .042 |
| | Ours | **.140** | .357 | **.233** | **.408** | **.206** | .426 | **.277** | **.434** | **.034** | **.042** | **.024** | **.057** | **.066** | **.037** |

Table 2: Average execution time (milliseconds / query) for each attack-model pair.

| | | MNIST | | | | | | | | CIFAR10 | | | | | |
|---|---|---|---|---|---|---|---|---|---|---|---|---|---|---|---|
| | | *Untargeted* | | | | *Targeted* | | | | *Untargeted* | | | *Targeted* | | |
| | Model | M1 | M2 | M3 | M4 | M1 | M2 | M3 | M4 | C1 | C2 | C3 | C1 | C2 | C3 |
| $\ell_0$ | BB | 10.76 | 11.85 | 10.19 | 12.02 | 60.88 | 62.17 | 62.31 | 57.74 | 46.51 | 50.31 | 50.43 | 99.71 | 105.28 | 103.53 |
| | Ours | **5.15** | **4.87** | **5.87** | **9.70** | **5.14** | **4.75** | **5.85** | **9.71** | **26.26** | **30.54** | **30.89** | **26.13** | **30.26** | **30.81** |
| $\ell_1$ | FAB | 9.38 | 8.88 | 12.61 | 36.00 | - | - | - | - | 84.04 | 108.91 | 108.64 | - | - | - |
| | BB | 6.73 | 7.03 | 7.31 | 12.50 | 43.25 | 43.54 | 43.69 | 43.86 | 32.56 | 37.40 | 37.59 | 68.99 | 73.33 | 74.03 |
| | Ours | **5.43** | **5.14** | **6.10** | **9.35** | **5.44** | **5.10** | **6.09** | **9.35** | **27.34** | **31.17** | **31.18** | **26.00** | **30.98** | **31.03** |
| $\ell_2$ | FAB | 10.22 | 10.13 | 13.45 | 36.72 | - | - | - | - | 84.27 | 109.43 | 108.87 | - | - | - |
| | CW | 4.22 | 4.09 | 5.17 | 10.07 | 4.23 | 4.14 | 5.15 | 10.06 | 25.90 | 31.32 | 31.31 | 25.78 | 31.32 | 31.30 |
| | BB | 4.44 | 4.15 | 5.03 | 12.38 | 26.20 | 26.76 | 27.24 | 31.00 | 26.64 | 31.82 | 31.90 | 48.74 | 54.35 | 54.07 |
| | DDN | **3.42** | **3.33** | **4.30** | **8.59** | **3.42** | **3.35** | **4.32** | **8.60** | **24.14** | **29.62** | **29.48** | **23.61** | **29.61** | **29.52** |
| | Ours | 4.46 | 4.42 | 5.48 | 9.15 | 4.50 | 4.44 | 5.47 | 9.09 | 24.88 | 30.22 | 30.08 | 25.39 | 30.21 | 30.04 |
| $\ell_\infty$ | FAB | 10.85 | 10.61 | 14.05 | 36.23 | - | - | - | - | 84.62 | 109.83 | 109.57 | - | - | - |
| | BB | 14.26 | 16.36 | 13.51 | 15.44 | 38.61 | 38.87 | 36.39 | 34.85 | 61.34 | 62.36 | 62.63 | 83.70 | 87.64 | 88.90 |
| | Ours | **4.25** | **4.33** | **5.30** | **9.17** | **4.33** | **4.23** | **5.31** | **9.10** | **24.84** | **30.15** | **30.01** | **24.78** | **30.19** | **30.03** |

attacks at fixed $\epsilon$ values are reported in Table 4. The results show that FMN outperforms or equals PGD in all norms.

## 4 Related Work

Gradient-based attacks on machine learning have a long history [3, 2]. Maximum-confidence attacks optimize the adversarial loss (e.g., the difference between the logits of the true class and the best non-true class) to find an adversarial point misclassifed with maximum confidence within a given, bounded perturbation size. While attacks in this category like FGSM [15], PGD [16, 17] and momentum-based extensions of PGD [26, 13] are popular, they only partially evaluate the adversarial

Table 3: Number of queries required by each attack to reach a perturbation size that is within 10% of the value obtained at $Q = 1000$.

|  | Model | MNIST | | | | | | | | CIFAR10 | | | | | |
|---|---|---|---|---|---|---|---|---|---|---|---|---|---|---|---|
|  |  | *Untargeted* | | | | *Targeted* | | | | *Untargeted* | | | *Targeted* | | |
|  | *Model* | M1 | M2 | M3 | M4 | M1 | M2 | M3 | M4 | C1 | C2 | C3 | C1 | C2 | C3 |
| $\ell_0$ | BB | **22** | **43** | 68 | **114** | 30 | 443 | 71 | 376 | 497 | 372 | 58 | 384 | 500 | 85 |
|  | Ours | **22** | 82 | 38 | 182 | **27** | **165** | **46** | **145** | **48** | **71** | **37** | **271** | **146** | **70** |
| $\ell_1$ | FAB | 44 | **242** | 152 | 569 | - | - | - | - | 124 | 220 | 72 | - | - | - |
|  | BB | 24 | 314 | 83 | **391** | 45 | 614 | 233 | 722 | 674 | 570 | 34 | 526 | 464 | 206 |
|  | Ours | **21** | 363 | **34** | 631 | **25** | **243** | **37** | **336** | **48** | **85** | **31** | **89** | **130** | **38** |
| $\ell_2$ | FAB | 14 | 60 | 40 | 532 | - | - | - | - | 18 | 28 | 14 | - | - | - |
|  | CW | 110 | 799 | 335 | - | 100 | 913 | 469 | - | 67 | 39 | 33 | 56 | 144 | 42 |
|  | BB | 20 | **24** | 20 | 337 | 21 | **61** | 20 | 692 | 22 | 23 | 22 | 26 | 27 | 29 |
|  | DDN | **12** | 136 | **15** | 474 | 12 | 149 | 26 | 670 | **13** | 20 | **4** | **18** | **19** | 18 |
|  | Ours | 16 | 94 | 16 | **190** | **11** | 136 | **16** | 188 | 28 | 23 | 7 | 25 | 29 | **13** |
| $\ell_\infty$ | FAB | 36 | 50 | 44 | 11 | - | - | - | - | 50 | 50 | 54 | - | - | - |
|  | BB | 19 | 17 | **20** | **5** | **24** | 17 | **22** | **5** | **20** | 24 | 21 | 27 | 33 | **29** |
|  | Ours | **9** | **10** | 22 | **5** | 27 | **8** | 26 | **5** | 22 | **15** | **14** | **20** | **29** | 34 |

robustness of a model. Minimum-norm attacks aim to minimize the norm of the perturbation subject to being adversarial. Attacks from this class give a more complete picture of the model robustness and allow us to compute the accuracy of the model under attacks with any post-hoc defined maximum perturbation size. L-BFGS [24] solves this problem with a quasi-Newton optimizer while CW [5] and EAD [8] use first-order gradient-based optimizers to minimize a weighted loss between perturbation size and misclassification confidence. To find the smallest adversarial perturbation, both CW and EAD need to tune the relative weighting which makes them query-inefficient. DeepFool [19] and SparseFool [18] compute gradients with respect to all classes in each step to estimate a linear approximation of the model from which the optimal adversarial perturbation can be computed. These two attacks are fast but do not converge to competitive solutions. BB [4] and FAB [10] use complex projections and approximations to stay close to the decision boundary (using the gradient to estimate the local geometry of the boundary) while minimizing the norm. This way of formulating minimum-norm optimization bypasses the tuning of a weighting term, but in the case of BB it also requires an adversarial starting point to begin with. The DDN attack [23] maximizes the adversarial criterion within a given norm constraint, and iteratively reduces the norm to find the smallest possible adversarial perturbation; however, it is constrained to $\ell_2$ and does not perform well on other $\ell_p$ norms.

The proposed FMN attack belongs to the category of minimum-norm attacks, and builds on BB, FAB and DDN to retain their main advantages. First, FMN is not specific to a given norm, and converges in many fewer steps than soft-constraint attacks like CW, as it does not need to optimize the trade-off between perturbation size and misclassification confidence. FMN needs significantly less computational time per step than the other attacks, it is very accurate and easy to use, and it does not necessarily require being initialized from an adversarial starting point.

Table 4: Success rate (%) of FMN against PGD on ImageNet models.

|  |  | ResNet18 | VGG |
|---|---|---|---|
| $\ell_1$ ($\epsilon = 1.0$) | PGD | 31.4 | 30.4 |
|  | FMN | **38.4** | **39.8** |
| $\ell_2$ ($\epsilon = 0.15$) | PGD | 61.7 | 61.4 |
|  | FMN | **65.8** | **66.2** |
| $\ell_\infty$ ($\epsilon = 4 \cdot 10^{-4}$) | PGD | 51.0 | **49.0** |
|  | FMN | **55.2** | **49.0** |

## 5   Contributions, Limitations, and Future Work

This work introduces a novel minimum-norm attack that combines all desirable traits to help improve current adversarial evaluations: (i) finding smaller or comparable minimum-norm perturbations across a range of models and datasets; (ii) being less sensitive to hyperparameter choices; and being extremely fast, by (iii) reducing runtime up to 3 times per query with respect to competing attacks and (iv) converging within less iterations. FMN also works with different $\ell_p$ norms ($p = 0, 1, 2, \infty$) and it does not necessarily require being initialized from an adversarial starting point. Our experiments have shown that FMN rivals or surpasses other attacks in speed, reliability, efficacy and versatility. While FMN is able to find smaller perturbations consistently when compared against $\ell_0$ and $\ell_1$ attacks, it only rivals the performance of other attacks for $\ell_2$ and $\ell_\infty$ norms, especially when tested against robust models which may present obfuscated gradients. To overcome this limitation, FMN may be extended using *smoothing* strategies that help find better descent directions, e.g., by averaging

gradients on randomly-perturbed inputs. This can be regarded as an interesting extension of FMN towards attacking robust models. In this respect, we also believe that FMN may facilitate minimum-norm adaptive evaluations in a more general sense. Adaptive evaluations, where the attack is modified to be maximally effective against a new defense, are the key element towards properly evaluating adversarial robustness [6, 25]. PGD attacks are popular in part for the ease by which they can be adapted to new defenses. Since FMN combines PGD with a dynamic minimization of the perturbation size, we argue that our attack can also be easily adapted to new defenses, thereby facilitating adaptive evaluations. FMN may also benefit from other improvements that have been suggested for PGD, including momentum, cyclical step sizes or restarts. We leave such improvements to future work.

To conclude, we firmly believe that FMN will establish itself as a useful tool in the arsenal of robustness evaluation. By facilitating more reliable robustness evaluations, we expect that FMN will foster advancements in the development of machine-learning models with improved robustness guarantees. We thus argue that there are neither ethical aspects nor evident future societal consequences with potential negative impacts that should be specifically addressed in the context of this work.

### Acknowledgements

This work has been partly supported by the PRIN 2017 project RexLearn (grant no. 2017TWNMH2), funded by the Italian Ministry of Education, University and Research; by the EU H2020 project ALOHA, under the European Union's Horizon 2020 research and innovation programme (grant no. 780788); and by BMK, BMDW, and the Province of Upper Austria in the frame of the COMET Programme managed by FFG in the COMET Module S3AI. Wieland Brendel acknowledges support from the German Federal Ministry of Education and Research (BMBF) through the Competence Center for Machine Learning (TUE.AI, FKZ 01IS18039A), from the German Science Foundation (DFG) under grant no. BR 6382/1-1 (Emmy Noether Program) as well as support by Open Philantropy and the Good Ventures Foundation.

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
