## Appendix

In this section we first show adversarial examples obtained by different $\ell_p$ attacks on MNIST and CIFAR10 data for visual comparison. These examples highlight the different behavior exhibited by each attack. We then report the query-distortion curves for all datasets, models and attacks used in this paper, showing that our attack outperforms current attacks on the $\ell_1$ norm and rivals their performance on other norms, while typically converging with much fewer queries.

### A1. Adversarial Examples

In Figs. 3-4, we report adversarial examples generated by all attacks against model M2 and C2, respectively, on MNIST and CIFAR10 datasets, in the untargeted scenario.

The clean samples and the original label are displayed in the first row of each figure. In the remaining rows we show the perturbed sample along with the predicted class and the corresponding norm of perturbation $\|\boldsymbol{\delta}^\star\|_p$. It is worth noting that the output class for different untargeted attacks is not always the same, which might sometimes explain differences in the perturbation sizes. An example is given in Fig. 4b, where the sample in the fourth column, labeled as "ship", is perturbed by most of the attacks towards the class "airplane", while in our case it outputs the class "dog" with a much smaller distance.

### A2. Query-distortion Curves

In Sect. 3.2 we introduced the query-distortion curves as an efficiency evaluation metric for the attacks. We report here the complete results for all models, in targeted and untargeted scenarios.

On the MNIST dataset, our attacks generally reach smaller norms with fewer queries, with the exception of M2 (Figs. 5-6), where it seems to reach convergence more slowly than BB in $\ell_0$ and $\ell_\infty$. In $\ell_2$, the CW attack is the slowest to converge, due to the need of carefully tuning the weighting term, as described in Sect. 4.

On the CIFAR10 dataset (Figs. 7-8), our attack always rivals or outperforms the others, with the notable exception of DDN for the $\ell_2$ norm, which sometimes finds smaller perturbations more quickly, as also shown in Table 3.

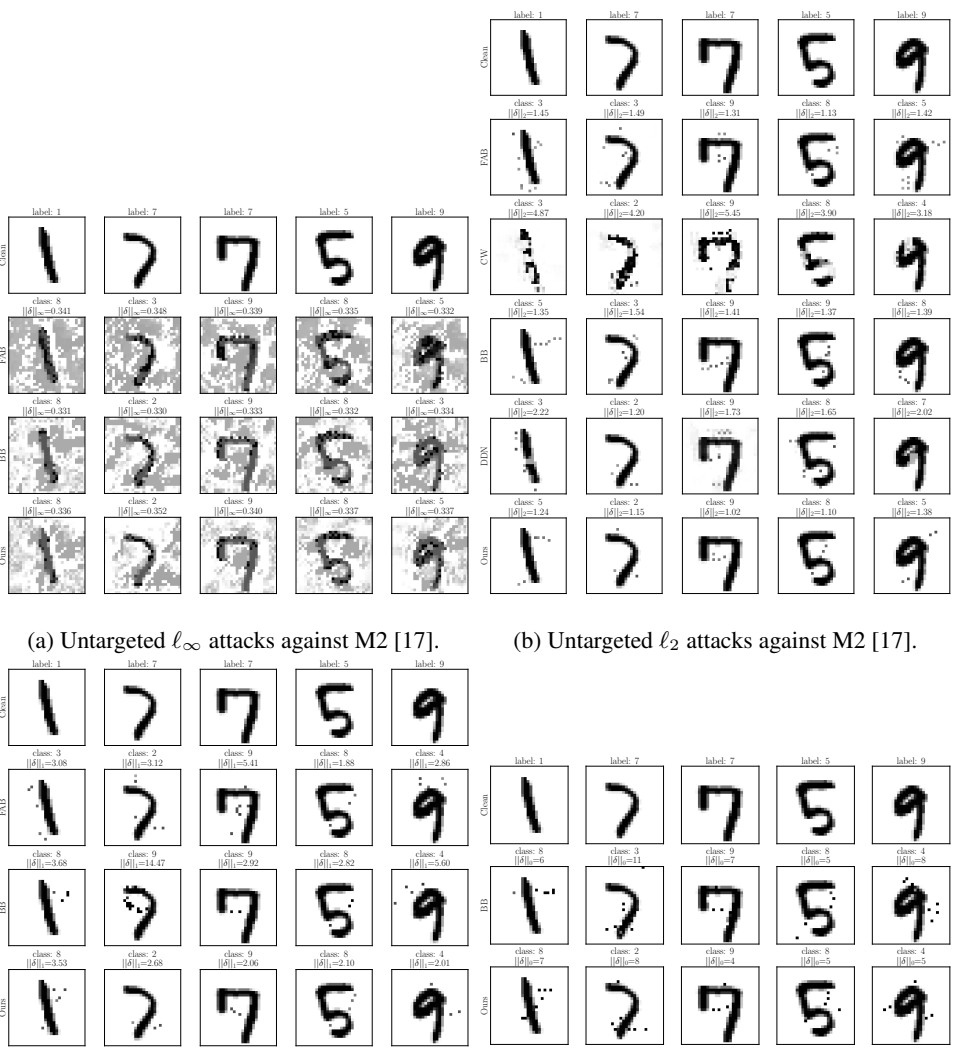

(a) Untargeted $\ell_\infty$ attacks against M2 [17].

(b) Untargeted $\ell_2$ attacks against M2 [17].

(c) Untargeted $\ell_1$ attacks against M2 [17].

(d) Untargeted $\ell_0$ attacks against M2 [17].

Figure 3: Adversarial examples on MNIST dataset.

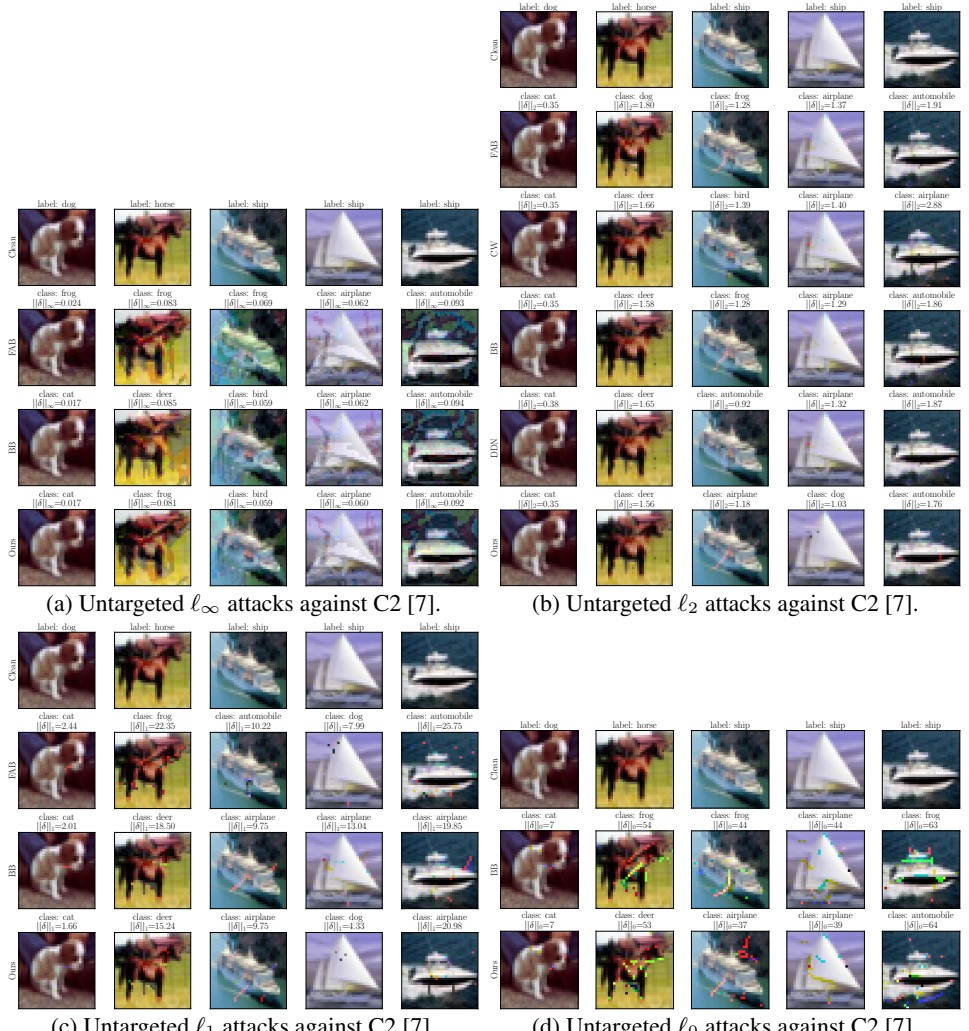

(a) Untargeted $\ell_\infty$ attacks against C2 [7].

(b) Untargeted $\ell_2$ attacks against C2 [7].

(c) Untargeted $\ell_1$ attacks against C2 [7].

(d) Untargeted $\ell_0$ attacks against C2 [7].

Figure 4: Adversarial examples on CIFAR10 dataset.

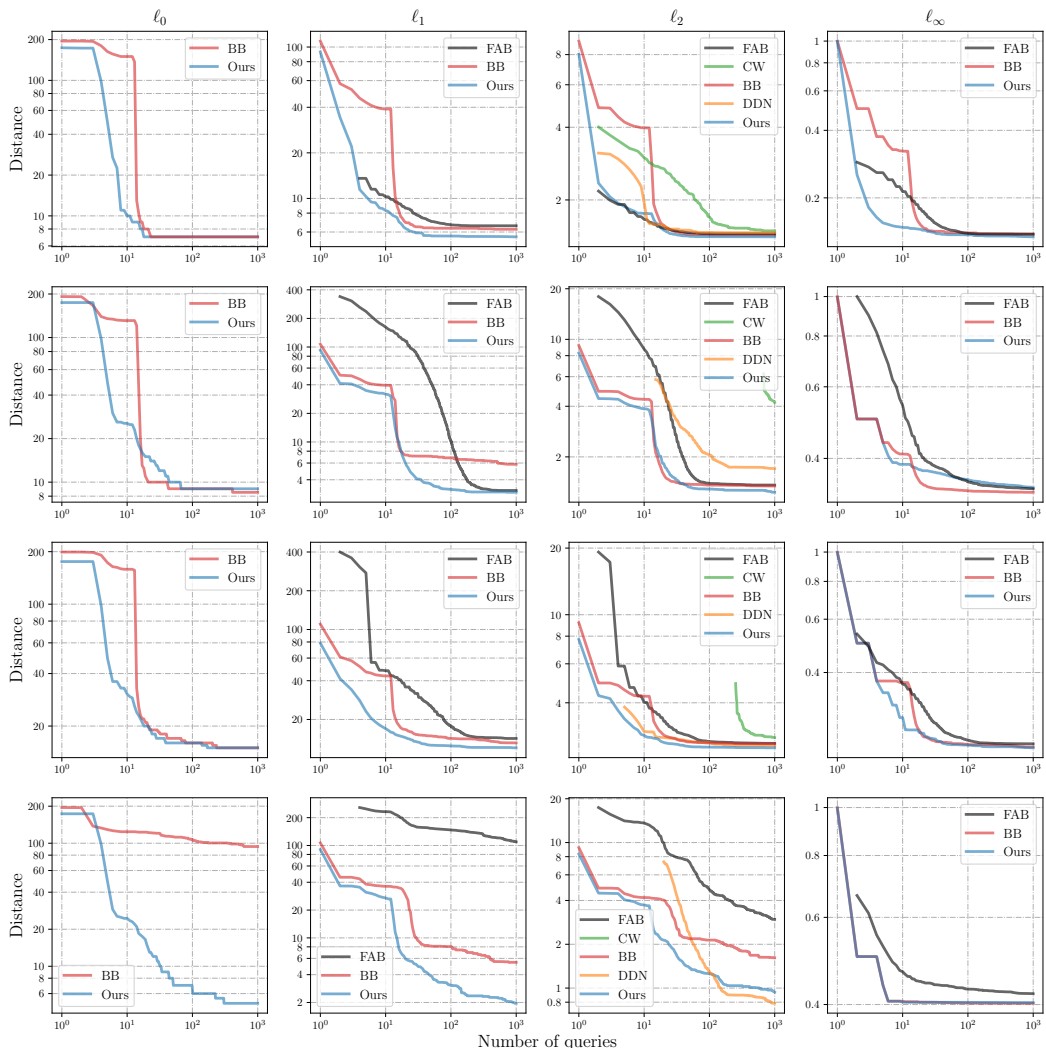

Figure 5: Query-distortion curves for untargeted ($U$) attacks on the M1, M2, M3, and M4 MNIST models.

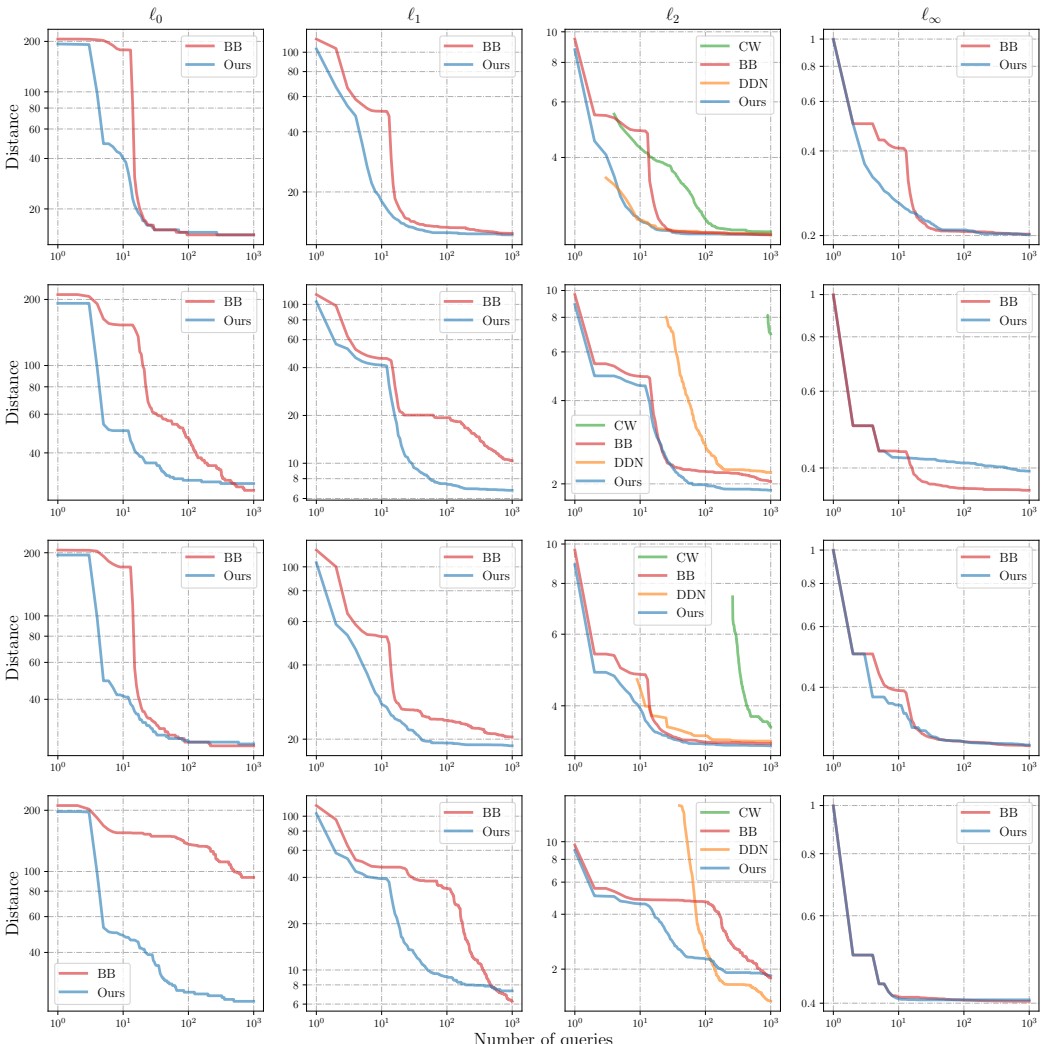

Figure 6: Query-distortion curves for targeted (*T*) attacks on the M1, M2, M3 and M4 MNIST models.

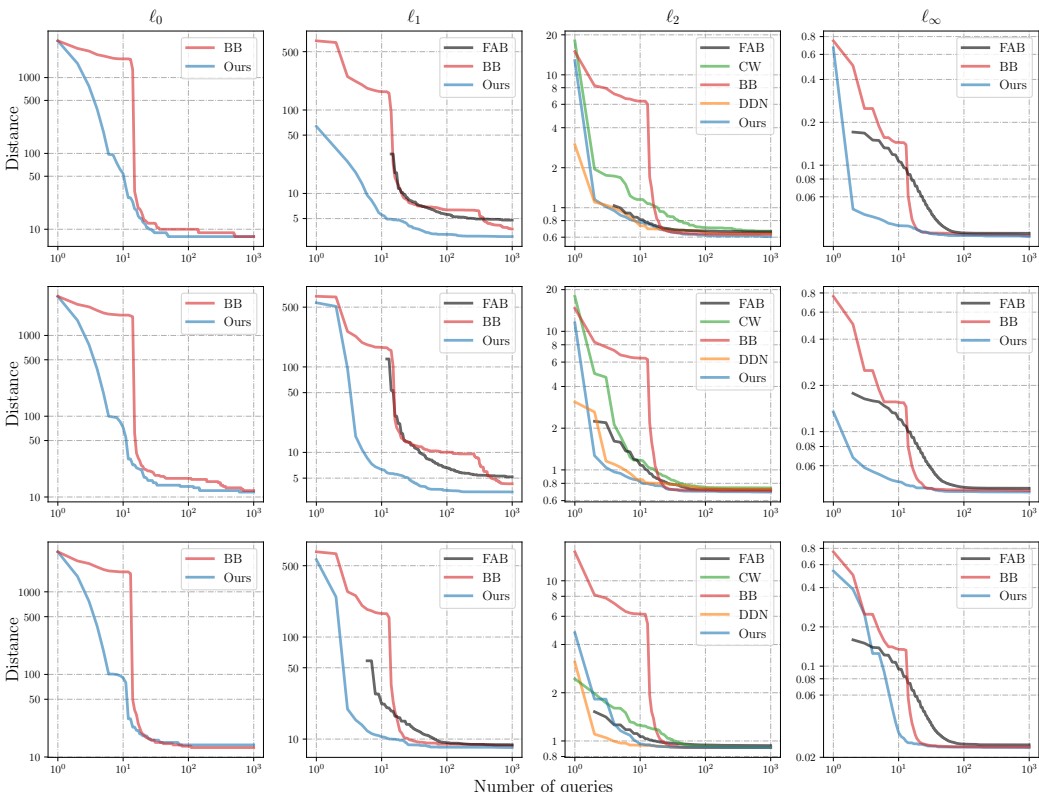

Figure 7: Query-distortion curves for untargeted (*U*) attacks on the C1 (*top*), C2 (*middle*), and C3 (*bottom*) CIFAR10 models.

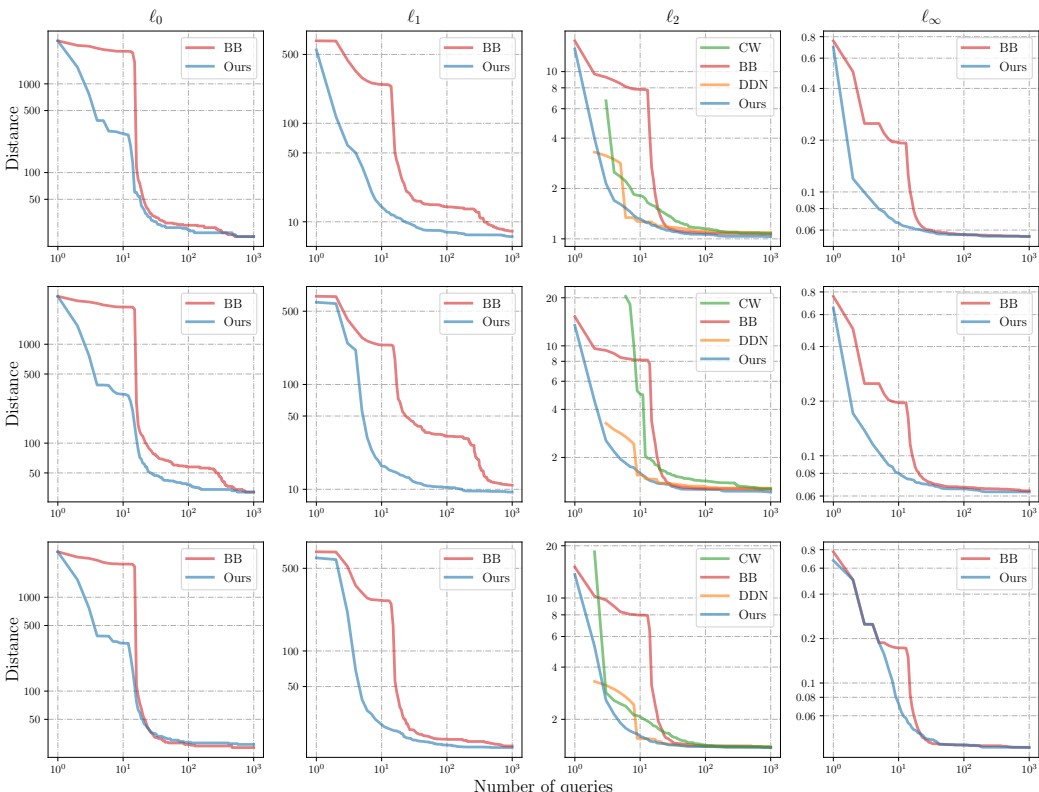

Figure 8: Query-distortion curves for targeted (*T*) attacks on the C1 (*top*), C2 (*middle*), and C3 (*bottom*) CIFAR10 models.