# OpenReview forum: "Fast Minimum-norm Adversarial Attacks through Adaptive Norm Constraints"
_NeurIPS.cc/2021/Conference — NeurIPS 2021 Poster_

### Official Review · Reviewer_Qt35 · 2021-07-10

**Rating:** 5
**Confidence:** 3

**Summary:**

This papers present new white-box minimum-norm adversarial attacks adapted to the case of the $\ell_p$ balls for $p=0,1,2,\infty$. It extends the DDN attacks which only considered $\ell_2$ norms.

**Ethical Concerns:**

None.

**Limitations And Societal Impact:**

Yes.

**Main Review:**

The experimental part is caried with much care and the paper is very well written and interesting to read. The main contribution of the paper is to extend a known technique that deal with $\ell_2$ constraints to deal with $\ell_p$ (for $p=0,1,2,\infty$). This contribution seems however not sufficiently original and motivated for such a conference as NeurIPS. I also found some unclear statements that I will now details.

- Extension to other norms. The main contribution of this paper is to extend a known technique DNN dealing with $\ell_2$ constraints to adversarial attacks that can deal with $\ell_p$ for $p=0,1,2,\infty$. Yet, it is not sufficiently motivated:
1) The motivation for using different norms is not sufficiently pointed out.
2) Among the chosen norms, why choosing only the $\ell_p$ for $p=0,1,2,\infty$? There are several papers considering other norms as a way to "having an arsenal of diverse attacks that can be adapted to specifics defense [...]", see, e.g., Wasserstein balls [W2019] or group-lasso [Xu2018].
3) Can your algorithm adapt to other norms? If so, what are the experimental performances?
4) Looking only at $\ell_p$ type of neighborhood is very reminiscent of one of the claims of the authors that "most of them [adversarial attacks] are just small variations of the same technique, make similar assumptions [...]". Indeed, we can, for instance, understand this choice of the $\ell_p$ norms for a measure of perturbation as such a similar assumption shared by most of works.


- Line 25-27 "While it may seem that the number of attacks is already large, most of them are just small variations of the same technique, make similar underlying assumptions and thus tend to fail jointly".
This sentence is much too vague and unprecise:
1) Can you defend this negative statement with a better explanation or references?
2) What type of "similar underlying assumptions" are you referring to?
3) Your sentence might be understood as a claim of a causal link between the failure of the methods and the fact that they make the same underlying assumptions. Can you write more explicitly that this is an observation rather than an established causal link? Or if not provide refs, explanations or numerical evidences?
4) To what same technique are you referring?
5) "Most of them" is an unprecised claim.


Some minor comments:
- Title: the title misleading let the reader believe that there is a great generality in the variety of different norms the paper deal with while it focuses only on the $\ell_p$ with $p=0,1,2,\infty$. So, could you either modify the title or explicit in the paper that your method is versatile to any norms?

- Line 96 "decayed with cosine annealing": It would be nice to add a reference.

- Line 127 FMN does not use the cross-entropy loss but rather the logit difference as the loss function $L$". Can you explain the rationale behind this difference with DNN?

[Xu 2018] Kaidi Xu, Sijia Liu, Pu Zhao, Pin-Yu Chen, Huan Zhang, Quanfu Fan, Deniz Erdogmus, Yanzhi Wang, and
Xue Lin. Structured adversarial attack: Towards general implementation and better interpretability

[W2019] Eric Wong, Frank R Schmidt, and J Zico Kolter. Wasserstein adversarial examples via projected sinkhorn
iterations.


**Time Spent Reviewing:**

6 hours.

---

> ### Author Response · Authors · 2021-08-09
> **Response to Reviewer Qt35**
>
> We would like to thank the reviewer for his/her comments, which we try to address below.
>
> **Extension to other norms is not sufficiently motivated.** To answer this comment, we would like to remark that the use of $\ell_1$, $\ell_2$, and $\ell_\infty$ norms has been considered in the majority of works in adversarial learning as an initial attempt to improve adversarial robustness against simple and tractable threat models (convex norms). Despite the simplicity of such attack models, the problem of learning robust models against them is still largely unsolved and challenging. The importance of the simpler attack models is also witnessed by the fact that the most popular adversarial libraries (e.g., Foolbox, Cleverhans), and benchmarks (e.g., RobustBench, MNIST Challenge, CIFAR Challenge) are almost entirely based on them. We do believe that FMN provides a substantial advancement in the development of proper testing tools to evaluate adversarial robustness on these norms, both fastly and reliably.
>
> **Common failures of attacks (lines 25-27).** As replied to reviewer sShH, our comments on commonalities and failures of attacks are mostly related to the categorization of attacks reported afterward in the introduction, which includes: (i) soft-constraint attacks (like CW), (ii) boundary attacks (like BB and FAB) and (iii) projected-gradient attacks (like PGD, MIM, BIM, etc.). While we quickly discussed pros and cons for each category in the Introduction, we agree that this may sound ambiguous and will thus rephrase such sentences to make it clearer that some categories of attacks may fail under some conditions (see, e.g., [1], in which projected-gradient attacks all fail in the same way against the “Ensemble Diversity” defense). It is not our goal to overstate our contributions, and we apologize if we gave this impression to the reviewers.
>
> We hope that our response clarifies the points raised by the reviewer and that the reviewer may reconsider them from a new perspective.

---

### Official Review · Reviewer_sShH · 2021-07-12

**Rating:** 7
**Confidence:** 5

**Summary:**

This paper presents an adversarial attack (FMN) that generates minimum-norm adversarial examples. The attack builds on the DDN attack but generalizes to $l_p$ norms beyond the $l_2$ norm that the DDN attack is specialized for.

- For the $l_0$, $l_1$ and $l_\infty$ norms, the attack has better performance: it has a consistently lower execution time per query, is more query efficient, and results in adversarial examples of comparable or smaller norm compared to the state-of-the-art.
- For the $l_2$ norm, the attack has comparable performance to the DDN attack: it has a slightly higher execution time per query, has comparable query efficiency, and results in adversarial examples of comparable norm compared to the state of the art (DDN).

In addition to the advantages in execution time and query efficiency, the method does not require adversarial starting points as some other methods do.

**Limitations And Societal Impact:**

Yes.

**Main Review:**

# Review Summary

## Strengths

- The paper has conducted a very thorough evaluation comparing the proposed attack to other attacks that are the state-of-the-art.
- The faster execution time for FMN means that it will be a useful tool for practitioners trying to estimate the minimum-norm attack on a classifier for a particular clean example.

## Areas of Improvement

- The experimental data provided does not substantiate the claim that FMN is more robust to the choice of hyperparameter.
  - The fact that FMN performs better when hyper-parameters are tuned at the dataset-level could simply mean that the best hyperparameter is the same across most of the samples (allowing, for example, the same hyperparameters to be applied to a test sample once hyperparameter tuning has been carried out on the training set), but does not mean that changing the hyperparameters at the dataset-level would not have a significant impact on attack performance.
  - Furthermore, for the $l_2$ and $l_\infty$ norms, the best alternative method (DDN and BB respectively) see a similar degradation in attack performance to FMN when comparing the sample-level and dataset-level hyperparameter tuning.
- As detailed in the section on originality, it is not clear what changes to the DDN algorithm enable FMN to generalize to norms other than $l_2$.
- As detailed in the section on clarity, this paper has some statements that are not supported by either references or any of the results presented, and would be greatly improved by clarifying these statements. One that I would like to highlight in particular is the claim in the abstract that "FMN significantly outperforms existing attacks in terms of convergence speed and computation time"; this is not borne out in the data for the $l_2$ norm. (The success of FMN on other norms is still valuable, and I'd rather this paper not overstate its contribution.)

---

# Originality

The presented FMN attack appears similar to the DDN attack [25] in many respects, such as the choice of how to decrease / increase $\epsilon$ and the projection and clipping step at the end of the for loop.

While there are differences in details that make for a substantially different algorithm (as identified at the end of Section 2), it would be helpful to understand which of the differences (if any) contribute to the performance improvement (e.g. via an ablation study). This is particularly important since DDN and FMN perform basically on par with each other on the $l_2$ examples shown. In addition, this paper claims in Line 44 that DDN "is specific to the $l_2$ norm and cannot be readily extended to other norms" --- is the issue that it is not possible to project $\delta_k$ onto an $\epsilon-k$ sphere in the appropriate norm, or is it simply that this leads to poor performance (perhaps this is what Lines 124-126)?

# Quality

This work presents thorough experimental results comparing FMN against state-of-the-art attacks, comparing the strength of the attacks as well as their runtime.

# Clarity

## Unsupported Statements

This work has some statements that are not supported by either references or any of the results presented. These statements should be clarified to avoid confusing readers with unsupported assertions, and ensure that the contributions of the paper can be clearly understood.

- Line 20: "all attacks make certain assumptions about the underlying geometry and properties of the optimization problem" -> Is there a reference to support this assertion? What assumptions are made about the geometry?
- Line 26: "most (attacks) ... are just small variations of the same technique, make similar underlying assumptions and thus tend to fail jointly" -> Is there a reference to support this assertion? In particular, which sets of attacks tend to fail jointly, and which are distinct?
- Line 39: "These attacks (like BB or FAB) ... may require an adversarial starting point" -> Do some of these attacks require an adversarial starting point, or do all of them require one sometimes?
- Line 50: "while finding equal or better optima" -> this phrasing suggests that FMN finds a successful adversarial attack of smaller norm on a per-sample basis (which _may_ be true; my objection is merely that the presented results do not demonstrate this); this paper should either show results to prove this, or emphasize that the claim is that the _average_ minimum norm found by FMN is smaller.
- Line 131: "initializing the attack from an adversarial point, which can greatly increase the convergence speed of the algorithm" -> while this may be true for other attacks (and makes intuitive sense), I could not find evidence in the paper to support the assertion that initializing attacks from an adversarial point increases convergence speed - much less "greatly".
- Line 160 + Footnote 1:
  - The footnote explains that FAB is not evaluated in the targeted case because it "implements a substantially different attack", but does not elaborate on why making a comparison to this different attack is not appropriate. It seems like the issue that this paper is alluding to is that the "targeted" FAB attack is actually a method to accelerate finding an untargeted adversarial example, and does not guarantee (despite the name) that the resulting adversarial example will be labelled as belonging to the target category by the classifier. If so, it would be very helpful to make this clear; a short explanation like "FAB does not support generating adversarial examples with a particular target label" should suffice.
- Line 234: "The reason may be that these robust models exhibit noisy gradients and flat regions around the clean input samples" -> M2, M3, M4 and C1, C2, C3 are all robust models - is there any evidence to suggest that some of them have noisier gradients / flatter regions around the clean input samples than others?

## Miscellaneous Issues

- Line 1: "Evaluating adversarial robustness amounts to finding the minimum perturbation needed to have an input sample misclassified" -> not necessarily; the robustness of defended models is often reported against attacks with a fixed budget; determining whether or not such an attack exists can be computationally less expensive than finding the minimum perturbation.
- Algorithm 1, Line 6: The variable $q$ is referenced here without being introduced (it is first introduced in Line 86)
- Algorithm 1, Line 14: normalizes $g$ by its $l_2$ norm, similar to DDN --- is this meant to be the $l_q$ norm or is it intended?
- Figure 2: It would be helpful to label which figures correspond to the M2 and C1 model.

## Suggestions

- Table 3: I'm finding it challenging to interpret the results presented in Table 3 as it relates to the efficiency of each attack. If the final perturbation size $\epsilon$ at Q=1000 is the same, then an attack with a smaller number of queries required to reach a perturbation size $\epsilon'$ within 10% of $d$ would be superior - but Table 1 shows that there _is_ significant variation in the value at Q=1000 itself. If Attack 1 makes slow but steady progress to a significantly smaller perturbation size at Q=1000, while Attack 2 makes quick progress, but plateauing at a larger perturbation size, most practitioners would probably prefer to select Attack 1. Personally, I find that the query-distortion curves already encapsulate this information well; if data in tabular form is still preferred, perhaps presenting the number of queries required by each attack to reach a perturbation size within 10% of _the best known value across all attacks_ would be better.
- While I appreciated the detailed experiments provided, providing all of the data in tabular form in the main body of the paper made it harder for me to understand. (I understand that a previous reviewer may have asked for all of the data). Summarizing the data (for example showing the ratio of average execution time of other attacks to that of FMN in a scatter plot) would make it easier for a reader to grasp the information.

# Additional Questions

- When an adversarial example has been previously found, it seems like the $\epsilon$-increase amount is selected such that the algorithm could test an $\epsilon$ value that is larger than the best known adversarial example. (In particular, we could have $\epsilon_{k+1} = \epsilon_{k-1}(1-\gamma_k)(1+\gamma_{k+1}) \leq \epsilon_{k-1}(1-\gamma_k)(1+\gamma_k) = \epsilon_{k-1}(1-\gamma_k^2) < \epsilon_{k-1}$, so we would have another $\epsilon$-increase if the true minimum value of $\epsilon$ is between $\epsilon_{k-1}(1-\gamma_k^2)$ and $\epsilon_{k-1}$.) This seems like it would waste queries; I am curious why a binary-search approach is not used instead.

---

# Updated Review

I am updating the score for this submission (6 -> 7) as the authors have effectively addressed my concerns (both in the "areas for improvement" and in "clarity") which were affecting my overall evaluation. Please see the commen thread below for specifics.

**Time Spent Reviewing:**

9

---

> ### Author Response · Authors · 2021-08-09
> **Response to Reviewer sShH**
>
> We would like to thank Reviewer sShH for the detailed comments and review of our work.
>
> **FMN robustness to hyperparameter choice.** In our evaluation, we consider an attack algorithm to be more robust to hyperparameter selection if the same set of hyperparameters works well on different samples, i.e., at the dataset level, as opposed to choosing the best hyperparameters for each sample. The reason is that the attack hyperparameters are often tuned at the dataset level in many practical cases when conducting robustness evaluations (rather than performing a complex, time-consuming grid search on each sample). Our experimental evaluation shows that many attacks indeed underperform in this setting, as reported in Table 1. We apologize if our definition of “robustness to hyperparameter selection” was not sufficiently clear in the paper, and we will definitely clarify what we mean by “robustness to hyperparameter selection” in the paper.
>
> **Comparison with DDN and contributions.** We applied significant improvements to DDN, as detailed in our paper and also acknowledged by the reviewer. It is however worth clarifying better here why DDN cannot be readily applied to other norms (e.g., using a different projection operator). The reason is that DDN rescales the perturbation to the value of $\epsilon$ at each iteration. This operation is based on the assumption that the loss stays linear (potentially also when $\epsilon$ is very large), and it causes problems when dealing with sparse norms, as it tends to move the perturbation in an $\ell_2$ direction (i.e., losing indeed sparsity). Of course, changing the rescaling step strategy required substantial changes to the whole optimization process. For instance, if the perturbation size is not equal to $\epsilon$ at each iteration, DDN ends up making smaller steps to reach the boundary, and this causes the update on the $\epsilon$ constraint to grow out of control ($\epsilon$ keeps increasing unless an adversarial example is found).
> This behavior has required many further modifications to the DDN algorithm, leading us to eventually develop the FMN attack.
>
> **Response to the unsupported claims.** We reply to the unsupported claims raised by the reviewer below, pointing out that we will rephrase and clarify most of these points in the paper.
> * Line 20-26: these comments are mostly related to the categorization of attacks reported afterward in the introduction: (i) soft-constraint attacks (like CW), (ii) boundary attacks (like BB and FAB), and (iii) projected-gradient attacks (like PGD, MIM, BIM, etc.). For each category, we quickly discussed pros and cons in the introduction, but we will rephrase the sentences in lines 20 and 26 to make it clearer that some categories of attacks may fail under some conditions (see, e.g., [1], in which projected-gradient attacks all fail in the same way against the "Ensemble Diversity" defense). It is not our goal to overstate our contributions, and we apologize if we gave this impression to the reviewers.
> * Line 39: BB requires the adversarial point to start, while FAB does not require adversarial points as initialization. We will clarify that in the introduction.
> * Line 50: We conducted a per-sample analysis, and indeed we confirm that FMN finds better optima in general, especially on $\ell_0$ and $\ell_1$. We will clarify this statement in the manuscript, including additional metrics to support the claim (e.g., average minimum norm).
> * Line 131: Our attack uses a strategy similar to BB to find the boundary between the two classes. While starting from the initial point requires performing several steps of gradient descent, initializing from the adversarial class allows using a line search to get closer to the boundary quicker. The line search uses only forward queries to the model, hence avoids computing gradients until the boundary is hit. The gradient is later used for refining the result and achieving a minimum-norm perturbation. This allows FMN to spare queries to the model and to converge quicker.
> * Footnote 1 + line 160: FAB lacks a targeted attack model as it is usually intended. Our purpose in writing this was to guide the reader in understanding why we did not include the FAB attack in the targeted case. We did not mean to discredit the FAB attack in any way. We will rephrase this part in the paper to make it clearer.
> * Line 234: The statement on the robust models exhibiting noisy gradients and flat regions gives a possible explanation; however, it is not strongly supported by empirical evidence in our work. An analysis of this phenomenon is given in [2]. We will anyway clarify this sentence in the paper.
>
> We hope to have answered the concerns of the reviewer on the claims that we made, and we will take care of updating the manuscript to clarify our statements.
>
> **Miscellaneous.** We thank the reviewer for pointing out such issues, which we will address in the revised manuscript. We would just like to clarify that in line 14 of Algorithm 1 we are seeking to preserve the direction of the gradient. For this reason, we divide for the $\ell_2$ norm of the vector and not by the dual norm $\ell_q$, which would instead project the sample along a different direction (e.g. the $\ell_\infty$ projection brings the vector to a corner of the ball, hence changing the direction of the original vector). We found this strategy to be particularly effective for sparse norms.
>
> **Additional Question.** The reviewer is right. However, since the problem is non-convex and has multiple optima, it is fine to let the algorithm explore larger balls in search of a better solution (potentially after a few iterations). In particular, allowing a small increase of epsilon enables the point to jump across the boundary in search of a better solution. Conversely, the algorithm would get stuck when hitting the boundary for the first time (see Fig. 1, middle, for a clear example of this dynamic). We thank the reviewer for digging so deep into our work, we hope that the explanation is clear enough, and will try to reflect it in the paper too.
>
>
> [1] Tramer, Florian, Nicholas Carlini, Wieland Brendel, e Aleksander Madry. «On Adaptive Attacks to Adversarial Example Defenses». http://arxiv.org/abs/2002.08347.
>
> [2] Liu, Chen, Mathieu Salzmann, Tao Lin, Ryota Tomioka, e Sabine Süsstrunk. «On the Loss Landscape of Adversarial Training: Identifying Challenges and How to Overcome Them». http://arxiv.org/abs/2006.08403.

---

> > ### Comment · Reviewer_sShH · 2021-08-27
> > **Question regarding l_2 norm performance + miscellaneous comments**
> >
> > Thank you for your response, and sorry for my late reply.
> >
> > I have one remaining major concern (outlined in the original review) that I would like to hear from the authors about before making a final decision on my scorer.
> >
> > Specifically, the claim in the abstract that "FMN significantly outperforms existing attacks in terms of convergence speed and computation time" does not seem to be true for the $l_2$ norm, with DDN having comparable performance. Am I misunderstanding the metrics presented? If not, the work should make it clear that FMN simply *matches* the state of the art for the $l_2$ norm and improves on it for other $l_p$ norms (still a significant achievement for the same technique to improve on all these other norms!)
> >
> > ---
> >
> > Here are my comments on the rest of the response.
> >
> > > **FMN robustness to hyperparameter choice.**
> >
> > Thanks for the explanation; this clears it up. Looking forward to seeing the explanation in the paper.
> >
> > > **Comparison with DDN and contributions.**
> >
> > This explanation was very helpful! I know you have limited space, but I think it would help the community to have a clear idea for why the changes to DDN that FMN implements are required. I hope that you will be able to integrate this explanation in the paper.
> >
> > > Line 20-26: ... It is not our goal to overstate our contributions, and we apologize if we gave this impression to the reviewers.
> >
> > I didn't think that you were trying to overstate your contributions relative to existing methods here - and I understand that you are working with limited space when trying to summarize the many different types of existing methods.
> >
> > I do think that it would be valuable for the community for these statements to be precise and properly cited, especially since a future researcher might quote your assertion! (For example, your citation for how PGD attacks fail in a similar way against "Ensemble Diversity" would be helpful).
> >
> > > Footnote 1 + line 160: ... We did not mean to discredit the FAB attack in any way.
> >
> > Again, I didn't think that you were trying to discredit the FAB attack here. My first impression when reading the paper was "Hey, I didn't know that FAB had a targeted attack, how does their targeted attack work?", and upon re-reading the FAB paper, I realized that the 'targeted' FAB attack is actually a method to accelerate finding an untargeted adversarial example, and that no 'targeted' FAB attack in the most common sense of the word exists. Thanks for rephrasing your footnote.
> >
> > > we are seeking to preserve the direction of the gradient.
> >
> > Would you be able to add a one-line comment in your pseudocode stating this? I spent some time trying to figure out why $l_2$ was the appropriate norm to use.
> >
> > > since the problem is non-convex and has multiple optima, it is fine to let the algorithm explore larger balls in search of a better solution
> >
> > I'm not sure where this explanation would go in the paper, but I think it would be helpful for it to be in the paper itself (rather than just on OpenReview) for future researchers.

---

> > > ### Author Response · Authors · 2021-08-31
> > > **Thanks for the feedback and further comments**
> > >
> > > This review and the feedback on our response helped us concretely pinpoint where the paper can be adjusted to significantly improve its clarity. For this, we are extremely grateful to the reviewer for the time and dedication put into the review of our work. We would like to describe how we would act on the paper, informed by the reviewer’s feedback.
> > >
> > > > Specifically, the claim in the abstract that "FMN significantly outperforms existing attacks in terms of convergence speed and computation time" does not seem to be true for the norm, with DDN having comparable performance. Am I misunderstanding the metrics presented? If not, the work should make it clear that FMN simply matches the state of the art for the norm and improves on it for other norms (still a significant achievement for the same technique to improve on all these other norms!)
> > >
> > > We thank the reviewer for giving us the opportunity to further clarify this point, we really appreciate it. We essentially agree with the given comment, and we will make it explicit in the abstract that FMN matches the performance of state-of-the-art attacks on the $\ell_2$ norm, while outperforming them on other norms. We will then clarify in the main text that FMN significantly outperforms DDN only against the defended model M2 on MNIST, where FMN converges with fewer queries to a value that is better than the asymptotic value reached by DDN at 1000 queries. Note also that in the majority of the remaining cases, FMN still improves over DDN, but only marginally.
> > >
> > > > I know you have limited space, but I think it would help the community to have a clear idea for why the changes to DDN that FMN implements are required.
> > >
> > > > Would you be able to add a one-line comment in your pseudocode stating this? I spent some time trying to figure out why L2 was the appropriate norm to use.
> > >
> > > We agree that including details on the algorithm and differences with DDN would help justify the choices that we made, and how the attack itself works. We will add a one-line explanation on the use of the $\ell_2$ norm in the pseudocode too, referring to it as ‘gradient-scaling step’, and clarifying in the text that it is intended to preserve the gradient direction before actually projecting the perturbation onto the epsilon-sized ball in the given norm.
> > >
> > > > I do think that it would be valuable for the community for these statements to be precise and properly cited, especially since a future researcher might quote your assertion! (For example, your citation for how PGD attacks fail in a similar way against "Ensemble Diversity" would be helpful).
> > >
> > > We thank the reviewer for highlighting the parts that were not clear or could be misunderstood by the readers. We will apply the suggested changes and make the statements less prone to ambiguity by citing the corresponding references. We will report the example on the Ensemble Diversity defense in the text, along with citing the corresponding reference.
> > >
> > > > I realized that the 'targeted' FAB attack is actually a method to accelerate finding an untargeted adversarial example, and that no 'targeted' FAB attack in the most common sense of the word exists. Thanks for rephrasing your footnote.
> > >
> > > We thank the reviewer for taking the time to investigate this issue. We did the same when looking for a targeted version of that attack. We will try to further clarify this point in the paper.

---

### Official Review · Reviewer_qsb1 · 2021-07-16

**Rating:** 8
**Confidence:** 5

**Summary:**

The paper proposes a new adversarial attack called Fast Minimum Norm (FMN) attack. FNM retains the major advantages of DDN (Decoupled direction and norm attack) [1]. While DDN is specific to the l_{2} norm and cannot be easily extended to other norms, the authors show that FMN can be directly extended to other norms such as l_{0}, l_{1} and l_{\infty} norms.

The authors conduct experiments using many different models and datasets and FMN outperforms previous attacks in terms of convergence speed. The minima found are either comparable or better across almost all tested scenarios.

I agree with the author's assessment that it can be a very useful tool for adversarial robustness evaluation.

[1] J. Rony, L. G. Hafemann, L. S. Oliveira, I. B. Ayed, R. Sabourin, and E. Granger. Decoupling direction and norm for efficient gradient-based l2 adversarial attacks and defenses.

**Limitations And Societal Impact:**

Since the paper proposes a new adversarial attack that can be used to comprehensively test robustness of a neural network against many different threat models, I do not foresee any potential negative societal consequences of this work.

**Main Review:**

The authors propose a new adversarial attack called FMN which keeps the advantages of the previously proposed DDN, while also allowing threat models with norms other than l_{2}.

This paper is clearly written and easy to understand.

Pros:

1. The experimental setup uses multiple different models and datasets (MNIST, CIFAR-10 and Imagenet). The evaluation is very comprehensive.

2. FMN has the additional advantage that it does not require the starting point of the optimization to be an adversarial example.

3. The proposed attack achieves smaller (or comparable) perturbation sizes with lesser computation time when compared to the existing state-of-the-art methods. It can be a very useful tool when carrying out robustness evaluations.

4. The proposed attack works well under different l_{p} norm threat models. Results are significantly better than the previous state of the art for the l_{0} and l_{1} threat models.

Cons:

1. The authors should clarify why they use the median as a measure of robustness instead of the widely used mean. But I did not let this negatively affect my review because median has been used in a previous NeurIPS publication [1].

Overall, I believe it is a solid contribution and recommend acceptance.


[1] W. Brendel, J. Rauber, M. Kümmerer, I. Ustyuzhaninov, and M. Bethge. Accurate, reliable and fast robustness evaluation. In Advances in Neural Information Processing Systems, pages 12861–12871, 2019.


**Time Spent Reviewing:**

8

---

> ### Author Response · Authors · 2021-08-09
> **Response to Reviewer qsb1**
>
> We thank Reviewer qsb1 for appreciating our work, and for remarking that our attack can provide another useful tool in adversarial robustness evaluations.
>
> We would just like to clarify that we used the median (as done in the previously-published NeurIPS paper) as a performance measure because it corresponds to the perturbation size for which the attack has a 50% success rate. We avoided using the mean as it would not have such a direct interpretation, and also as it would be only defined when all testing points are misclassified (if an attack fails, the distance for that given point is set to infinity in our experiments), and normally achieving 100% success rate only happens when $\epsilon$ is large and not even for all attacks.

---

### Official Review · Reviewer_Rii9 · 2021-07-18

**Rating:** 5
**Confidence:** 4

**Summary:**

This paper proposes a new adversarial attack algorithm for finding the smallest perturbation necessary to fool a classifier under various $\ell_p$ norms. The algorithm is similar to DDN, but includes a few improvements, including using the logit margin loss instead of cross-entropy, using a more complex algorithm for updating the bound $\epsilon$, and not always projecting the adversarial example to the boundary. They evaluate the new attack, FMN, on a variety of classifiers on MNIST and CIFAR-10.


**Limitations And Societal Impact:**

Limitations and societal impact were adequately addressed.

**Main Review:**

First, I would like to commend the authors on the clear presentation and extensive experiments in the paper. I found the paper easy to read, with the algorithm and experiments clearly organized and explained. The separation of the sample-level and dataset-level hyperparameter tuning was helpful to compare typical attack use (dataset-level tuning) and best-case performance. The authors were also clear about the relationship with prior work.

The novelty of FMN is somewhat limited, since it is quite similar to DDN. Accordingly, the results for FMN are quite similar to DDN, although they seem to generally improve on it slightly. The applicability of FMN to the $\ell_0$ and $\ell_1$ norms is useful. Since the differences between FMN and DDN are heuristics and the results empirical, it would improve the paper if an ablation study were done to show which changes between DDN and FMN actually lead to the improved performance of FMN. This could lead to more general insights about how to improve adversarial attacks.

Overall, while the paper is well-presented, the results are somewhat marginal and it is not clear which changes in FMN lead to the improved performance over DDN.

Potential typos/grammatical issues:
 * Line 72: should "and" be "added"?
 * Line 86: "being *q*" should be "*q* being"
 * Line 298: "much" should be "many"

**Time Spent Reviewing:**

1.5

---

> ### Author Response · Authors · 2021-08-09
> **Response to Reviewer Rii9**
>
> We would like to thank Reviewer Rii9 for appreciating our work (in particular, for acknowledging that its presentation is clear and the experiments are extensive), and for providing constructive comments. We respond below to some of the statements given in the review.
>
> **Limited novelty with respect to DDN.** Even if our attack is similar in spirit to DDN, DDN only works for $\ell_2$, while FMN works effectively also for $\ell_0$, $\ell_1$, and $\ell_\infty$ norms (in particular, FMN does much better than the competing attacks especially on $\ell_0$ and $\ell_1$ norms).
> We do believe that this is a valuable contribution in terms of novelty already on its own, as the aforementioned extension is non-trivial, and also considering the wide experimental analysis conducted in our study. In addition, DDN is substantially outperformed by FMN in cases like M2, in both targeted and untargeted cases, where FMN not only reaches better results (Table 1) but also faster convergence (Figure 2).
>
> To enable extending the ideas used by DDN to other norms, FMN improves the initial formulation of DDN in many aspects, as detailed in the last paragraph of Sect. 2:
> 1. FMN does not always rescale the perturbation size to be equal to $\epsilon$ (which hinders the optimization process for $\ell_0$ and $\ell_1$ attacks);
> 2. FMN does not optimize the cross-entropy loss, as it may exhibit saturation issues/noisy gradients when used against defended models (see, e.g., [1] and M2 in our work), but it rather uses the logit difference (as suggested in [1]);
> 3. FMN does not require an initial estimate of the $\epsilon$ distance (i.e., DDN has an additional hyperparameter to be tuned), and it also uses an improved linear approximation to estimate $\epsilon$ at each iteration, which is less likely to overshoot the minimum in cases where gradient obfuscation is present or where the loss landscape has a large curvature;
> 4. FMN decays $\gamma$ to avoid oscillations around the minimum; and
> 5. FMN allows initializing the attack from the adversarial class, which can speed up convergence (as shown in many query-distortion curves, FMN starts finding adversarial points much earlier).
>
> **Insights on how to improve adversarial attacks.** Based on the discussion above, even if we have not considered a detailed ablation study as suggested by the reviewer (also due to space constraints and considering an already-massive experimental analysis), we still believe that many useful lessons can be drawn from our paper to understand how current adversarial attacks can be improved. In particular, we overcome the limitations found in DDN by formalizing an attack that (1) is defined in $\ell_0$, $\ell_1$, $\ell_2$, and $\ell_\infty$ norm; (2) uses a loss that suffers less from saturation effects; (3) relies only on local linear approximations of the loss landscape; (4) removes a hyperparameter difficult to estimate a priori, especially for different norms; (5) further refines the results with decay on the step for the $\epsilon$-update; and (6) speeds up convergence by a smart adversarial initialization. We will clarify these aspects in the paper.
>
> **Final remarks.** For these reasons, we believe that our contributions are not marginal and our work will have a large impact on the community. In particular, having a fast algorithm that works as well as (or even better than) DDN on norms different than $\ell_2$ would certainly help perform better adversarial robustness evaluations and develop better defense algorithms in a much larger number of cases (other than just $\ell_2$). We hope that the reviewer will appreciate our response.
>
> [1] Carlini, Nicholas, e David Wagner. «Defensive Distillation is Not Robust to Adversarial Examples». http://arxiv.org/abs/1607.04311.

---

### Author Response · Authors · 2021-08-09
**Summary of the reviews and main response**

Dear Reviewers and Area Chair,
We summarize here the main points that were raised by the reviewers and how we answered them.

**Novelty with respect to DDN.** Reviewers Rii9, Qt35, and sShH are concerned that this work seems incremental with respect to DDN, even though DDN only works on $\ell_2$. While we agree that the underlying idea of alternating a maximum-confidence step to an epsilon-tuning step is the same used by DDN, the main contribution of this paper is indeed to extend this idea to develop fast minimum-norm attacks also beyond $\ell_2$. This extension has not been trivial, as we had to make different changes to the initial DDN implementation to make it work effectively also on other norms (especially sparse ones). In particular, we needed to carefully consider gradient normalization and projection to adapt the steps made in DDN to other norms. Considering different $\ell_p$ projections and using DDN ‘as is’ simply did not work. We had to rethink the whole algorithm, as one may appreciate by comparing the algorithm reported in our paper and that reported in the DDN work. Moreover, our experimental evidence shows that, for the majority of the models considered, FMN outperforms the other attacks (including DDN) by perturbation size and query efficiency, especially in the more practical cases in which adversarial robustness is evaluated by keeping the attack hyperparameters fixed for each sample (i.e., dataset-level hyperparameter tuning).

**Importance of considering other norms.** Reviewer Qt35 questions the importance of extending DDN to other norms, requiring us to motivate why considering $\ell_0$, $\ell_1$, $\ell_2$, and $\ell_\infty$ norms is important. Adversarial robustness has been mostly studied on such $\ell_p$ norms, as witnessed by the number of attack and defense papers published using such threat models, and also by popular benchmarks like RobustBench. The reason is that finding mitigations even to simpler, more tractable attacks (e.g., using convex norms) has proven extremely challenging, hindering the research progress in this area. Despite different threat models are also available (e.g., semantic and optimal-transport attacks), still, most of the defenses try to improve robustness against the aforementioned $\ell_p$ norms; thus, we do believe that FMN may constitute a very useful tool not only to improve and speed up current adversarial robustness evaluations, but also to help design better defenses.

**Clarification on robustness to hyperparameter selection.** We have clarified in response to Reviewer that by “robustness to hyperparameter selection” we mean the robustness exhibited by the attack when selecting the hyperparameters at the dataset level (i.e., in the most commonly-used setting for adversarial robustness evaluations) rather than on a sample basis. This means that the selected hyperparameter configuration works well across different samples, which is different from measuring the sensitivity of the algorithm on the same sample when changing the hyperparameters. We will clarify this aspect in the paper.

**Unsupported claims.** We agree with Reviewers sShH and Qt35 on the need to better substantiate or rephrase some claims in the paper. We have replied in the point-to-point responses below, and we will also update the paper accordingly.

We thank the reviewers again for their detailed and constructive comments and hope they can re-evaluate our paper as informed by our responses.

---

> ### Comment · Reviewer_sShH · 2021-08-27
> **Importance of considering $l_p$ norms.**
>
> I agree with the authors that considering $l_p$ norms is valuable for the reasons outlined. I'd also like to add that techniques for $l_p$ robustness can be used to work with perturbation sets where the perturbation cannot be mathematically defined, such as image corruptions and adversarial lighting variations [1]. This strengthens the case for working with $l_p$ norms.
>
> [1] Wong, Eric, and J. Zico Kolter. "Learning perturbation sets for robust machine learning." arXiv preprint arXiv:2007.08450 (2020).

---

### Decision · Program_Chairs · 2021-09-27

**Decision:**

Accept (Poster)

**Comment:**

The paper proposes a fast minimum-norm (FMN) attack that works with different norm perturbation models and is robust to hyper-parameter choices. Some reviewers had concerns regarding the novelty of the work with respect to prior works (e.g. DDN). Authors clarified some of these concerns in the discussion period. Overall I think the paper makes good contributions. I suggest authors to take reviewers' suggestions into account in the final draft of their work.